# A CDK1 phosphorylation site on *Drosophila* PAR-3 regulates neuroblast polarisation and sensory organ formation

**Nicolas Loyer[1], Elizabeth KJ Hogg[2], Hayley G Shaw[1], Anna Pasztor[1,2], David H Murray[1], Greg M Findlay[1,2], Jens Januschke[1]\***

[1]Molecular, Cell and Developmental Biology, University of Dundee, Dundee, United Kingdom; [2]MRC PPU, School of Life Sciences, University of Dundee, Dundee, United Kingdom

**Abstract** The generation of distinct cell fates during development depends on asymmetric cell division of progenitor cells. In the central and peripheral nervous system of *Drosophila*, progenitor cells respectively called neuroblasts or sensory organ precursors use PAR polarity during mitosis to control cell fate determination in their daughter cells. How polarity and the cell cycle are coupled, and how the cell cycle machinery regulates PAR protein function and cell fate determination is poorly understood. Here, we generate an analog sensitive allele of CDK1 and reveal that its partial inhibition weakens but does not abolish apical polarity in embryonic and larval neuroblasts and leads to defects in polarisation of fate determinants. We describe a novel in vivo phosphorylation of Bazooka, the *Drosophila* homolog of PAR-3, on Serine180, a consensus CDK phosphorylation site. In some tissular contexts, phosphorylation of Serine180 occurs in asymmetrically dividing cells but not in their symmetrically dividing neighbours. In neuroblasts, Serine180 phosphomutants disrupt the timing of basal polarisation. Serine180 phosphomutants also affect the specification and binary cell fate determination of sensory organ precursors as well as Baz localisation during their asymmetric cell divisions. Finally, we show that CDK1 phosphorylates Serine-S180 and an equivalent Serine on human PAR-3 in vitro.

**\*For correspondence:**
j.januschke@dundee.ac.uk

**Competing interest:** The authors declare that no competing interests exist.

## Editor's evaluation

The coupling of polarity to the cell cycle is critical to ensuring polarization, spindle position and fate asymmetries are properly linked to cell cycle progression. Using a combination of an analog sensitive Cdk1 allele and phoshomimetic/non-phosphorylatable mutants, this important work convincingly shows the impact of Cdk1 on polarity domain coalescence, Baz/Par3 localisation, and fate specification that are of broad interest to the field.

## Introduction

The coordination of cellular events in time and space during cell division is important for development. This spatiotemporal organization is particularly critical for certain types of asymmetric cell division that generate different cell fates in a single division. Cell fate differences in the resulting daughter cells can stem, for instance, from the asymmetric subcellular localisation of fate determinants in the dividing cell. This asymmetry results in the daughter cell receiving distinct sets of molecular information. In such divisions, the subcellular localisation and segregation of molecules involved in fate decisions needs to be timed along the cell cycle for proper establishment of cells fates of the resulting daughter cells. The molecular mechanisms underpinning this coordination are not fully resolved.

Studies in worms and in flies have uncovered key aspects into the coupling of the cell cycle and cell fate and suggest that the cell cycle dependent regulation of cell polarity is important in this context (*Noatynska et al., 2013*; *Fichelson et al., 2005*). In many asymmetric dividing cells, the conserved PAR polarity complex, composed of PAR-3, PAR-6 and atypical protein kinase C (aPKC) (*Goldstein and Macara, 2007*; *Goehring, 2014*), polarises prior to division. The PAR polarity complex exhibits cell-cycle dependent localisation in the *C. elegans* zygote (*Kemphues, 2000*) and many stem and precursor cells, such as neural precursors in the chick (*Das and Storey, 2012*) and fish (*Alexandre et al., 2010*), muscle stem cells in the mouse (*Dumont et al., 2015*), and stem and precursor cells of the developing central and peripheral nervous system of *Drosophila* (*Wodarz et al., 1999*; *Schober et al., 1999*; *Bellaïche et al., 2001*). The mechanisms controlling the subcellular localisation and function of the PAR complex in cycling cells are poorly understood.

In the *C. elegans* zygote, the coordination of cell polarity by the cell cycle machinery is critical for a successful asymmetric division establishing the precursor of the germ line and that of somatic fates in a single division (*Rose and Kemphues, 1998*). Cell cycle kinases like Plk1 and Aurora A (*Kim and Griffin, 2020*; *Reich et al., 2019*; *Schumacher et al., 1998*) regulate polarity in the zygote required for the establishment of PAR anterior-posterior polarisation. In flies, the homologs of Plk1 and Aurora A and the master regulator of cell cycle transitions, CDK1 (*Nurse, 1997*), have been implicated in linking cell cycle and asymmetric division in the central and peripheral nervous system (*Tio et al., 2001*; *Wirtz-Peitz et al., 2008*; *Lee et al., 2006*; *Wang et al., 2006*; *Wang et al., 2007*).

*Drosophila* neural stem cells called neuroblasts are a well-studied model for asymmetric cell division (*Sunchu and Cabernard, 2020*; *Tassan and Kubiak, 2017*). Neuroblasts divide asymmetrically generating a self-renewed neuroblast and a differentiating daughter cell. This outcome relies on a series of events taking place during mitosis. In prophase, PAR proteins (PAR-6, aPKC and PAR-3, known as Baz in *Drosophila*) accumulate to a cortical domain, defining the apical pole (*Rolls et al., 2003*; *Wodarz et al., 1999*; *Wodarz et al., 2000*; *Petronczki and Knoblich, 2001*). The earliest signs of neuroblast polarisation in mitosis include the recruitment of Baz from the cytoplasm to a broad apical area at the onset of prophase (*Hannaford et al., 2018*). At nuclear envelope breakdown (NEB), basal-to-apical actin-driven cortical flows then lead to the coalescence of Baz into continuous a continuous cluster that forms a bright apical crescent (*Oon and Prehoda, 2019*). The signals that drive Baz coalescence and its functional significance in neuroblasts are unknown.

Following nuclear envelope break down, the PAR complex excludes cell fate determinants such as Miranda, Pon and Numb from the apical pole *via* aPKC activity, leading to their localisation at the opposite basal pole. At metaphase, the mitotic spindle aligns with the apical basal polarity axis, leading to the asymmetric segregation of fate determinants into the differentiating daughter cell (*Gillies and Cabernard, 2011*). Therefore, cell polarity and the cell cycle are tightly coordinated in neuroblasts, with apical polarity assembled and disassembled at each cell cycle. Baz is heavily regulated by phosphorylation by different kinases including PAR-1 and aPKC in flies (*Benton and St Johnston, 2003*; *Morais-de-Sá et al., 2010*) and additionally by Aurora A, Plk1 and other kinases in other systems (*Dickinson et al., 2017*; *Khazaei and Püschel, 2009*). Thus, cell-cycle-dependent kinases, indispensable for the spatio-temporal regulation of cell divisions, are promising candidates for coordinating polarity with the cell cycle through direct phosphorylation of Baz.

Some polarity proteins are phosphorylated by cell-cycle-dependent kinases during asymmetric neuroblast division: Aurora A phosphorylates PAR-6 (*Wirtz-Peitz et al., 2008*) and Polo, the fly homolog of Plk1, phosphorylates Pon (*Wang et al., 2007*). Importantly, although these phosphorylation events control basal polarity, apical Baz polarity remains largely unaffected in *aurora A* and *polo* mutants (*Lee et al., 2006*; *Wang et al., 2006*; *Wang et al., 2007*). Interestingly, Baz asymmetric localisation was reported to be lost in embryonic neuroblasts expressing a dominant negative form of CDK1 (*Tio et al., 2001*). Thereby, direct phosphorylation of Baz by CDK1 may be the temporal trigger driving Baz polarisation in mitotic neuroblasts.

Like in neuroblasts, Baz polarity is coupled to the cell cycle in asymmetrically dividing sensory organ precursors (SOPs) in the peripheral nervous system of the fly (*Bellaïche et al., 2001*). In this system, SOP cells are first specified through a Notch-dependent lateral inhibition mechanism (*Simpson, 1990*), and then undergo a series of asymmetric cell divisions generating five cells. At the end of each division, the asymmetric segregation of Notch regulators results in the differential activation of the Notch pathway in each daughter cell, assigning them different identities (*Schweisguth,*

*2015*). Downregulation of CDK1 activity in SOPs delays mitosis and causes mother-to-daughter cell fate transformations occurring without divisions, affecting polarity orientation when cells eventually divide, and ultimately leads to loss of external organs (*Fichelson and Gho, 2004*). Interestingly, Cyclin A, a CDK1 activator, was recently found to colocalise with Baz at the posterior cortex of mitotic SOPs (*Darnat et al., 2022*), consistent with the possibility that CDK1 may directly phosphorylate Baz.

Here, we used live cell imaging and chemical genetics to study the cell cycle and cell polarity coordination in *Drosophila* neuroblasts and SOPs. We generated an analog-sensitive allele of CDK1 allowing us to tune CDK1 activity. Partial CDK1 inhibition in larval neuroblasts affects Baz polarity by preventing apical crescent coalescence at NEB. We identified a consensus CDK phosphorylation on the Baz sequence (Baz-S180), against which we developed a phospho-specific antibody. In larval brains and the pupal notum at 16 hr after puparium formation (APF), Baz-S180 is phosphorylated specifically during the asymmetric division of neuroblasts and sensory organ precursors but not during the symmetric cell divisions of neighbouring cells. Baz-S180 is also phosphorylated in non-mitotic cells earlier in the pupal notum (8 hr APF), during the specification of sensory organ precursors. We generated Baz-S180 phosphomutants and observed that, despite not being necessary for Baz localisation, phosphorylation of Baz-S180 controls the timing of basal polarisation in neuroblasts. In sensory organs, Baz-S180 phosphomutants affect the specification of sensory organs precursors, binary cell fate decisions following their ACD, and Baz localisation during ACD of the pIIa cell. Finally, we report that Baz-S180 and an 'equivalent' Serine on human PARD3 are substrates for CDK1/Cyclin B in vitro.

## Results

### Generation and in vivo analysis of an analog-sensitive allele of CDK1

To investigate the role of CDK1 in regulating Baz localisation during the neuroblast cell cycle we generated an analog sensitive allele (*Lopez et al., 2014*) of *cdk1*, *cdk1^F80A^*, henceforth called *cdk1^as2^*. To test whether CDK1 activity can be acutely and specifically inhibited in these mutants, we exposed control and *cdk1^as2^* larval brains to the ATP analog 1-NA-PP1, using cell cycle arrest as an indicator (*Nurse, 1997*). As we previously showed that 10 µM of 1-NA-PP1 acutely blocks an analog sensitive version of aPKC (*Hannaford et al., 2019*), we used 10 µM as a starting concentration. While control neuroblasts continued proliferating after exposure to 1-NA-PP1, *cdk1^as2^* neuroblasts stopped dividing (*Figure 1A and A'*). Interestingly, upon 1-NA-PP1 addition, early prophase *cdk1^as2^* neuroblasts lost their apical Baz crescent and did not further proceed through mitosis (*Figure 1B*), apparently reverting back to interphase, a reported effect of CDK1 inhibition (*Potapova et al., 2011*). Another expected effect of CDK1 inhibition is exit from mitosis in metaphase-arrested cells (mitotic exit; *Potapova et al., 2006*). We first exposed neuroblasts to Colcemid to arrest them in metaphase, and then to 1-NA-PP1, which resulted in mitotic slippage within 10 min in the majority of *cdk1^as2^* neuroblasts, but not in controls (*Figure 1C and C'*). Interestingly, in neuroblasts that exited mitosis, Baz asymmetry was lost in a manner reminiscent of localisation changes normally occurring with the onset of anaphases of untreated neuroblasts. Like in anaphase, Baz crescents spread along the lateral cortex while the cell elongated along the apico-basal axis, after which Baz eventually dissipated into the cytoplasm (*Figure 1D*). We conclude that CDK1 activity is acutely and specifically inhibited in *cdk1^as2^* mutant flies by 10 µM of 1-NA-PP1.

### Partial inhibition of CDK1 prevents coalescence of apical Baz crescents

We next investigated whether acute CDK1 inhibition affects neuroblasts polarity. CDK1 inhibition with 10 µM 1-NA-PP1 prevents neuroblasts from cycling and causes metaphase-arrested neuroblasts to slip out of mitosis (*Figure 1*), which prevents the analysis of polarizing or polarized neuroblasts. Therefore, we partially inhibited Cdk1 using lower concentration of 1-NA-PP1. *cdk1^as2^* neuroblasts stopped proliferating when exposed to 1 µM or higher concentrations but continued proliferating at a slower rate in the presence of 0.5 µM (*Figure 1—figure supplement 1A*). Hence, 10 µM 1-NA-PP1 likely fully inhibits CDK1 and 0.5 µM partially inhibits CDK1. We compared apical polarity in mitosis of control and *cdk1^as2^* neuroblasts before and after exposure to 0.5 µM 1-NA-PP1. In control cells, Baz polarized in a dynamic manner. First, it was recruited to a broad cortical crescent during prophase, which then coalesced into a smaller, brighter crescent at NEB, as previously described for unperturbed neuroblasts (*Hannaford et al., 2018*; *Oon and Prehoda, 2019*). Interestingly,

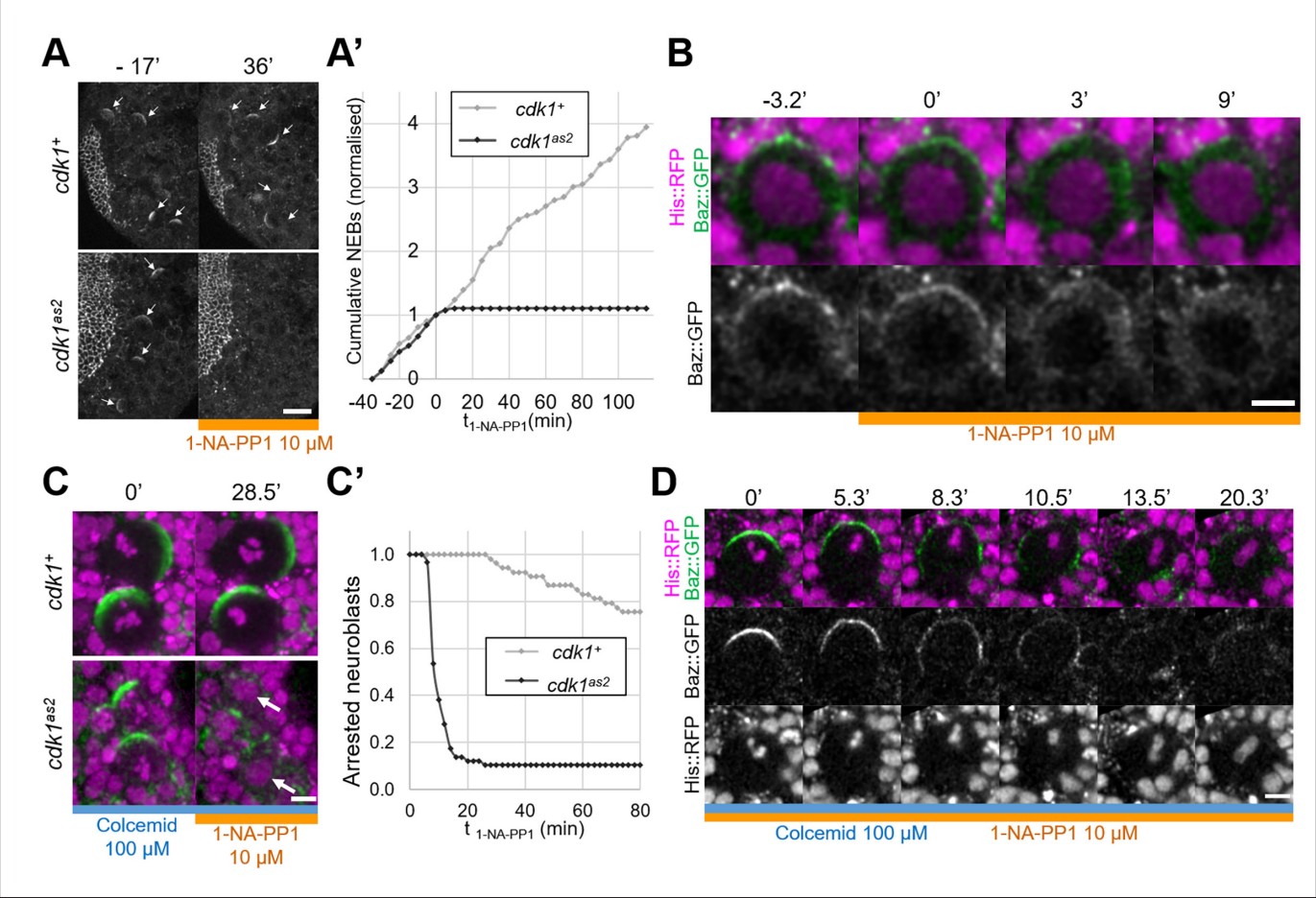

**Figure 1.** Full inhibition of analog-sensitive CDK1. (**A**) Live larval brains expressing Baz::GFP, before and after addition of 1-NA-PP1 10 μM at t0. Arrows point to mitotic neuroblasts. Scale bar: 20 μm. (**A'**) Cumulative sum chart of NEB events, normalized to the total number by the time of 1-NA-PP1 addition (**t0**). N=324 divisions in 4 brains (*cdk1⁺*) and 75 divisions in 5 brains (*cdk1ᵃˢ²*), 3 experiments. (**B**) Loss of Baz::GFP cortical localisation in a cycling *cdk1ᵃˢ²* neuroblast at prophase, after addition of 1-NA-PP1 10 μM at t0. Observed in 12 cases across 8 experiments. Scale bar: 5 μm. (**C**) Live neuroblasts metaphase-arrested by exposure to Colcemid 100 μM, expressing Baz::GFP (green) and His::RFP (magenta), before and after addition of 1-NA-PP1 10 μM at t0. Scale bar: 5 μm. Arrows: decondensed DNA after neuroblasts exit mitosis. (**C'**) Number of metaphase-arrested neuroblasts, normalised to their number at the time of 1-NA-PP1 10 μM addition (**t0**). N=53 neuroblasts in 5 brains (*cdk1⁺*) and 58 neuroblasts in 5 brains (*cdk1ᵃˢ²*), 2 experiments. (**D**) Mitotic slippage of a metaphase-arrested (Colcemid 100 μM) *cdk1ᵃˢ²* neuroblast after addition of 1-NA-PP1 10 μM at t0. Scale bar: 5 μm.

The online version of this article includes the following figure supplement(s) for figure 1:

**Figure supplement 1.** Titrating 1-NAPP1 concentration for partial CDK1 inhibition.

---

although Baz still polarized upon partial CDK1 inhibition in *cdk1ᵃˢ²* neuroblasts, apical Baz crescents appeared fainter and wider than during the previous cell cycle occurring before exposure to 1-NA-PP1 (*Figure 2A and A'*, *Figure 1—figure supplement 1B*). Thus, Baz crescents fail to coalesce at NEB upon partial CDK1 inhibition. As Baz polarity was previously shown to be completely lost in dominant negative *cdk1* mutant embryonic neuroblasts (*Tio et al., 2001*), we tested the effect of partial inhibition of CDK1 in larval neuroblasts in a sensitized context. Baz localisation is dependent on an oligomerization domain (OD) and a lipid binding domain (LD), which are functionally redundant (*Kullmann and Krahn, 2018*). We reasoned that deletion of either of these domains might make Baz localisation more sensitive to disruptions and lead to complete loss of polarity upon partial inhibition of CDK1. However, Baz deletion mutants for either of these domains still polarized when we partially inhibited CDK1, and only presented an apical coalescence defect (*Figure 2B*). To test whether embryonic neuroblasts are more sensitive to CDK1 inhibition than larval neuroblasts, we exposed embryonic neuroblasts to 0.5 μM 1-NA-PP1. Again, partial inhibition of CDK1 in embryonic

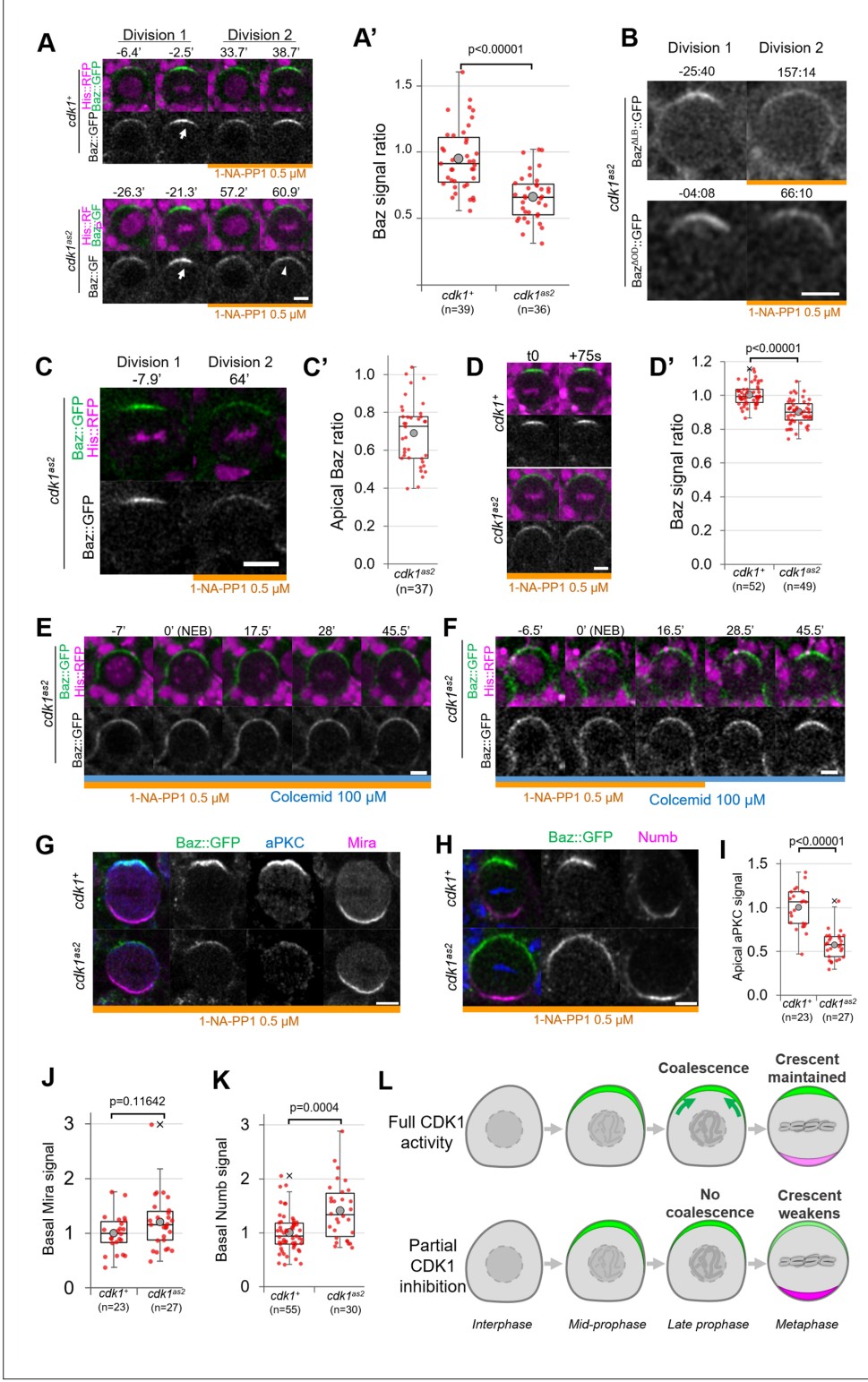

**Figure 2.** Partial inhibition of analog-sensitive CDK1. (**A**) Two consecutive divisions of live control and *cdk1^as2*
cycling neuroblasts expressing Baz::GFP, before and after addition of 1-NA-PP1 0.5 µM at t0. Arrow: coalesced
Baz crescent in metaphase. Arrowhead: non-coalesced crescent. Scale bar: 5 µm. (**A'**) Ratio between the apical
Baz signal at metaphase in the presence of 1-NA-PP1 0.5 µM and the Baz signal during the previous metaphase,
in the absence of 1-NA-PP1. In control neuroblasts, 1-NA-PP1 0.5 µM addition does not reduce the intensity of
Baz crescents compared to the previous cell cycle (loss of 5.3 ± 24%, n=39). In *cdk1^as2* neuroblasts, 1-NA-PP1

*Figure 2 continued on next page*

*Figure 2 continued*

addition reduces the intensity of Baz crescents compared to the previous cell cycle (loss of 34.2 ± 17%, n=36). Statistical test: two-tailed Mann–Whitney U test. For this boxplot and every following boxplot: cross: maximal and/ or minimal outliers (beyond 1.5×interquartile range); grey circle: average; red dots: individual measurements; centre line, median; box limits, upper and lower quartiles; whiskers, 1.5×interquartile range. 5 brains per condition across 3 experiments. (**B**) Two consecutive divisions of cycling *baz^815-8^*, *cdk1^as2^* neuroblasts expressing Baz^ΔLB^::GFP or Baz^ΔOD^::GFP, before and after exposure to 1-NA-PP1 0.5 μM. Scale bar: 5 μm. n=92 divisions in the presence of 1-NA-PP1 in 7 brains (Baz^ΔLB^) and 75 divisions in 5 brains (Baz^ΔOD^), 2 experiments. (**C**) Two consecutive divisions of cycling *cdk1^as2^* embryonic neuroblasts expressing Baz::GFP and His::RFP, before and after exposure to 1-NA-PP1 0.5 μM. (**C'**) Ratio between the apical Baz signal at metaphase in the presence of 1-NA-PP1 0.5 μM and the Baz signal during the previous metaphase, in the absence of 1-NA-PP1 0.5 μM in *cdk1^as2^* embryonic neuroblasts. Statistical test: two-tailed Mann–Whitney U test. n=37 successive divisions in 31 neuroblasts across 5 embryos, 3 experiments. (**D**) Cycling neuroblasts at metaphase. Scale bar: 5 μm. (**D'**) Control cycling neuroblasts maintain stable Baz levels throughout metaphase (+0.4 ± 6.1% in 75 seconds, n=52 neuroblasts, 4 brains) in the presence of 1-NA-PP1 0.5 μM. In CDK1^as2^ cycling neuroblasts, Baz levels decrease during metaphase in the presence 1-NA-PP1 0.5 μM (–9.8 ± 7.3% in 75 s, n=49, 6 brains). Statistical test: two-tailed Mann–Whitney U test. Two experiments. (**E**) *cdk1^as2^* neuroblast first treated with 1-NA-PP1 0.5 μM for 1 hr, and then treated with Colcemid 100 μM. Neuroblasts stay polarized during metaphase (Baz crescents were maintained in 40/40 neuroblasts arrested in metaphase for at least 45', 6 brains, 2 experiments). t0: NEB. Scale bar: 5 μm. (**F**) *cdk1^as2^* neuroblast first treated with 1-NA-PP1 0.5 μM for 1 hr, then treated with Colcemid 100 μM for 30 min, after which 1-NA-PP1 was washed out (in this case 16.5' after NEB). Some metaphase-arrested neuroblasts undergo Baz crescents coalescence following 1-NA-PP1 washout (n=19/51 neuroblasts, 7 brains, 3 experiments). t0: NEB. Scale bar: 5 μm. (**G**) Cycling neuroblasts fixed at metaphase, exposed to 1-NA-PP1 0.5 μM, expressing Baz::GFP (green) and stained for aPKC (blue) and Miranda (magenta). Scale bar: 5 μm. (**H**) Cycling neuroblasts fixed at metaphase, exposed to 1-NA-PP1 0.5 μM, expressing Baz::GFP (green) and stained for Numb (magenta). Scale bar: 5 μm. (**I**) Apical aPKC signal and (**J**) basal Mira signal in metaphase neuroblasts exposed to 1-NA-PP1 0.5 μM. n=23 neuroblasts in 5 brains (*cdk1^+^*) and 27 neuroblasts in 4 brains (*cdk1^as2^*), 2 experiments. (**K**) Basal Numb signal in metaphase neuroblasts exposed to 1-NA-PP1 0.5 μM. n=55 neuroblasts in 6 brains (*cdk1^+^*) and 30 neuroblasts in 13 brains (*cdk1^as2^*), 2 experiments. Statistical test: two-tailed Mann–Whitney U test. (**L**) In control neuroblasts, an apical Baz crescent (green) assembles in prophase and coalesces (green arrows) into a narrower, brighter crescent at NEB. Its intensity is maintained throughout metaphase. Upon partial inhibition of CDK1, an apical Baz crescent (green) assembles in prophase but fails to coalesce at NEB and its intensity decreases during metaphase. Basal polarity proteins (magenta) form more intense crescents than in controls.

neuroblasts resulted in defective coalescence but not in complete loss of polarity (*Figure 2C*). We conclude that high levels of CDK1 activity are needed for full apical polarization of embryonic and larval neuroblasts by driving apical coalescence.

## Partial inhibition of CDK1 destabilizes Baz crescents in cycling neuroblasts

Upon partial CDK1 inhibition, we also observed that the non-coalesced Baz crescents of cycling *cdk1^as2^* neuroblasts displayed a small but significant decrease in intensity comparing two consecutive time points 75 s apart during metaphase (–9.8 ± 7.3%), whereas Baz crescent intensity in control neuroblasts was maintained (*Figure 2D and D'*), suggesting a role of CDK1 in maintaining Baz polarity. To investigate this possibility, we allowed neuroblasts to cycle into Colcemid-induced metaphase arrest under partial CDK1 inhibition. We reasoned that experimentally increasing the duration of metaphase would give enough time for unstable Baz crescents to completely disappear. However, Baz crescents remained stable in metaphase-arrested *cdk1^as2^* neuroblasts exposed to 0.5 μM 1-NA-PP1 (*Figure 2E*). The observation that non-coalesced crescents remain stable in metaphase enabled us to next test whether restoring normal CDK1 activity in metaphase-arrested neuroblasts could reinstate Baz crescent coalescence. Indeed, 37% of metaphase-arrested neuroblasts with non-coalesced crescents gradually concentrated Baz in narrow and brighter crescents following 1-NA-PP1 washout (*Figure 2F*). Thus, partial inhibition of CDK1 can briefly destabilize Baz crescents in cycling neuroblasts, but not in metaphase-arrested neuroblasts. Furthermore, NEB and Baz coalescence can be uncoupled by temporally interfering with CDK1 activity.

## Partial inhibition of CDK1 leads to non-coalescence of apical aPKC crescents and to increased basal Numb levels

The function of Baz coalescence during neuroblast polarization is unknown. As one key function of Baz is the recruitment of aPKC, unsurprisingly, aPKC also localised in dimmer apical crescents upon partial CDK1 inhibition (*Figure 2G, I*). As aPKC orchestrates basal cell fate determinant localisation by directly phosphorylating them or their adapters, we tested whether weaker aPKC polarity was correlated with abnormal basal polarization. Indeed, Numb basal crescents were significantly (41 ± 51%) more intense in *cdk1^{as2}* neuroblasts compared to controls under partial CDK1 inhibition (*Figure 2H and K*). Miranda crescents also displayed a slight, statistically insignificant increase (20 ± 49%) in intensity upon CDK1 partial inhibition (*Figure 2G and J*). In conclusion, partial inhibition of CDK1 prevents the coalescence of apical polarity crescents and leads to increased levels of basal fate determinants (*Figure 2L*).

## Baz is specifically phosphorylated in asymmetrically dividing cells of the central and peripheral nervous system on Serine180, a consensus CDK phosphorylation site

To determine whether the observed effects of CDK1 inhibition on Baz localisation are direct, we interrogated the Baz amino acid sequence for full consensus CDK phosphorylation sites [ST]Px[KR] (*Songyang et al., 1994*). We identified three phosphorylation sites, among which two have been reported to be phosphorylated in phosphoproteomic approaches in *Drosophila*: Serine S180 (*Bodenmiller et al., 2008*; *Zhai et al., 2008*; *Hilger et al., 2009*) and Serine 417 (*Reich et al., 2019*). The presence of a Cyclin-binding motif [RK]XL(X)[FYLIVMP], (*Lowe et al., 2002*) and of phospho-dependent Cks1-binding motifs [FILPVWY]XTP, (*McGrath et al., 2013*) further suggests the possibility of CDK regulation of Baz (*Figure 3A*). Given its proximity to both a Cyclin-binding motif and a Cks1-binding motif, we focused our analysis on Serine 180 and generated a phosphospecific antibody (anti-Baz-pS180), which we first used on larval brains after trichloroacetic acid (TCA) fixation (*Hayashi et al., 1999*). Remarkably, a Baz-pS180 signal was only detected in dividing neuroblasts, and neither in interphase neuroblasts nor in neighboring neuroepithelial, regardless of their cell cycle stage (*Figure 3B and C*).

This difference in Baz-S180 phosphorylation between asymmetrically dividing mesenchymal neuroblasts and symmetrically dividing neuroepithelial cells may arise from differences in cell division modes and/or cell type. To test this, we stained Baz-pS180 in the pupal notum at 16 hr APF, where we could compare symmetric and asymmetric cell divisions of epithelial cells. Again, mitotic asymmetrically dividing sensory organ precursors (SOPs) were positive for Baz-pS180, whereas symmetrically dividing epidermal cells were not (*Figure 3D*). Similarly, in late L3 larval imaginal disks, dividing SOPs were positive for Baz-pS180 while symmetrically dividing columnar epithelial cells were not (*Figure 3—figure supplement 1A, B*). Interestingly, the CDK1 interacting partner CyclinA was recently found to localise to the apical-posterior cortex of dividing SOPs (*Darnat et al., 2022*). We confirmed that cortical CyclinA colocalises with Baz-pS180 in mitotic SOPs (*Figure 3—figure supplement 1*), potentially providing a mechanism for this asymmetric cell division-specific phosphorylation event. This prompted us to examine the localisation of CDK1-associated Cyclins in neuroblasts. However, neither Cyclin A, Cyclin B nor Cyclin B3 localised to the cortex of dividing neuroblasts, indicating that neuroblast-specific phosphorylation of Baz-S180 is not achieved through cortical Cyclin localisation in the larval brain (*Figure 3—figure supplement 1*). We conclude that at least in the larval brain, late L3 larval imaginal discs and the pupal notum at 16 hr APF, Baz-S180 is apparently specifically phosphorylated during asymmetric cell divisions.

## Phosphorylation of Baz-S180 affects the timing of neuroblast basal polarization

Having established that Baz-S180 is phosphorylated in vivo, we next tested its functional relevance in larval neuroblasts. To this end, we generated UASz-driven (*DeLuca and Spradling, 2018*) constructs containing either wildtype (Baz^{WT}::GFP), non-phosphorylatable (Baz^{S180A}::GFP) or phosphomimetic (Baz^{S180D}::GFP) GFP-tagged versions of Baz. To assess the function of these constructs in the absence of endogenous Baz, we used UAS-driven RNAi against RFP to deplete a functional Baz::mScarlet-I CRISPR knock-in that we generated previously (*Houssin et al., 2021*). This resulted in an apparently complete depletion of Baz::mScarlet and expected Mira localisation defects (*Atwood et al., 2007*; *Atwood*

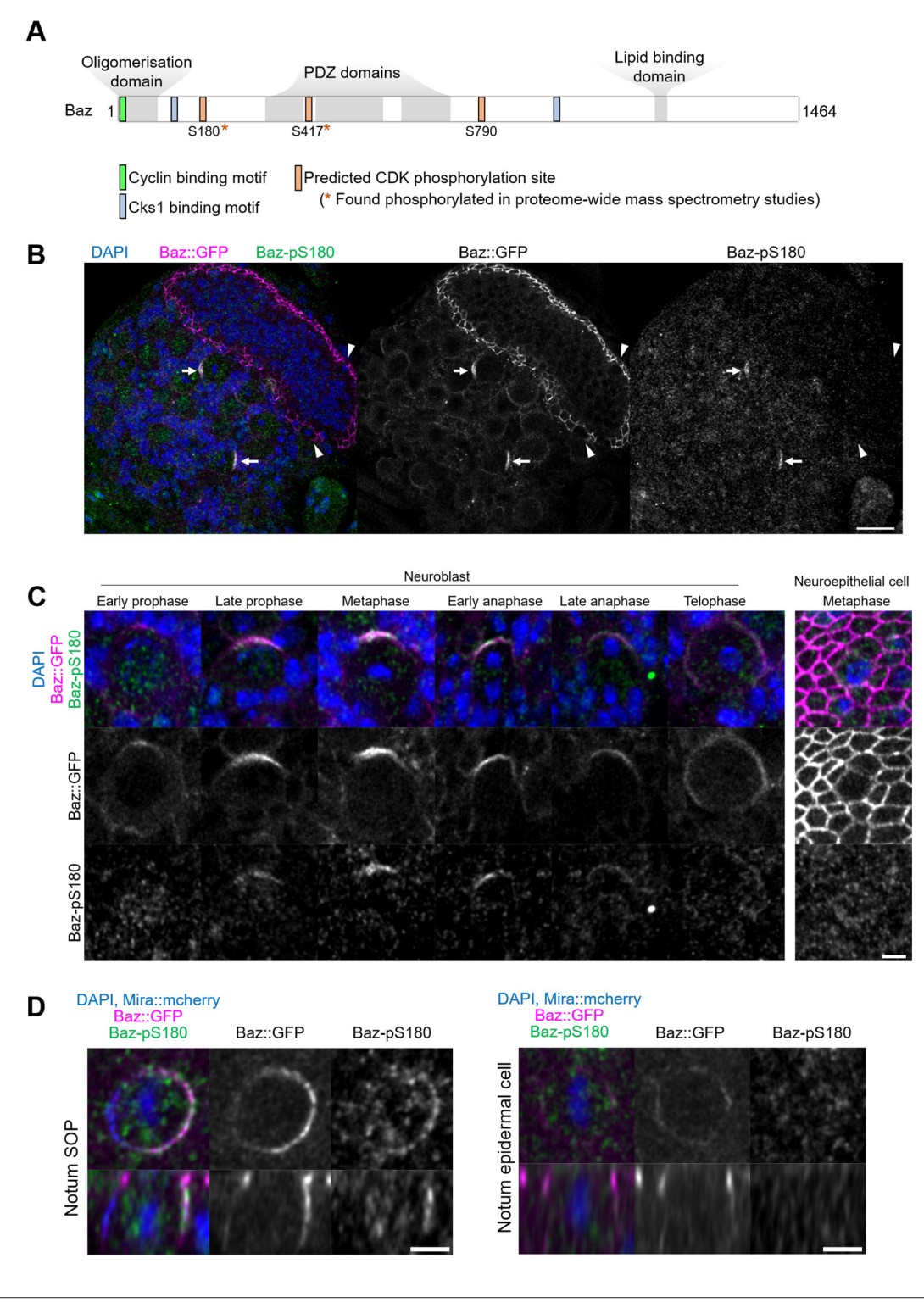

**Figure 3.** Asymmetric cell division-specific phosphorylation of Baz-S180. (**A**) Features of the Baz protein. Consensus motifs suggesting a potential phosphoregulation of Baz by CDK1 are highlighted. (**B**) Fixed fly brain expressing Baz::GFP (magenta) and stained with a phospho-specific antibody against Baz-pS180 (green). Arrows: mitotic neuroblasts. Arrowheads: mitotic neuroepithelial cells. Scale bar: 20 μm. (**C**) Left set of panels: cell cycle stages of fixed neuroblasts. Right set of panels: metaphase neuroepithelial cell. Scale bar: 5 μm. A Baz-pS180 signal was observed in 118/118 metaphase neuroblasts and 0/31 metaphase neuroepithelial cells (13 brains, 3 experiments). (**D**) Left: notum microchaete sensory organ precursor in metaphase, at 16 hr APF. Right: notum epidermal cell in metaphase, at 16 hr APF. Bottom panels: orthogonal view. Scale bar: 5 μm.

*Figure 3 continued on next page*

*Figure 3 continued*

The online version of this article includes the following figure supplement(s) for figure 3:

**Figure supplement 1.** Localisation of Cyclins in asymmetrically dividing cells and Baz-S180 phosphorylation in imaginal discs.

*and Prehoda, 2009*), which were rescued by expression of our Baz^WT::GFP construct (*Figure 4A*). We proceeded to investigate the localisation and function of our Baz-S180 phosphomutant constructs in Baz::mScarlet-I-depleted neuroblasts. First, we examined their localisation in cycling neuroblasts and observed that both phosphomutant constructs seemingly localised similar to Baz^WT::GFP throughout the cell cycle (*Figure 4B and C*). All constructs were likewise stably retained at the apical pole in metaphase-arrested neuroblasts (*Figure 4D*). We also took advantage of the Baz^S180A::GFP construct to test the specificity of the Baz-pS180 antibody. In the absence of endogenous Baz, Baz^WT::GFP apical crescents were positive for Baz-pS180 whereas Baz^S180A::GFP apical crescents were not, confirming the specificity of the antibody (*Figure 3—figure supplement 1*).

We next assessed the functionality of Baz-S180 phosphomutants constructs in establishing basal polarity. In both phosphomutants, we detected a significant delay in the formation of basal Mira crescents. These crescents reached an equivalent cortical/cytoplasmic ratio to Baz^WT::GFP-expressing neuroblasts only at metaphase in Baz^S180A::GFP and anaphase in Baz^S180D::GFP (*Figure 4E and F*). Numb localises to basal crescents in metaphase in all Baz^WT::GFP-expressing, all Baz^S180A::GFP-expressing and most Baz^S180D::GFP-expressing neuroblasts. Interestingly, in Baz^S180D::GFP-expressing neuroblasts where Mira showed very low polarization (cortical/cytoplasmic ratio <2), Numb often (n=8/16 neuro-blasts) localised uniformly to the cortex (*Figure 4G and G'*). Like for Mira, this Numb mislocalisation in metaphase was only transient as Numb localised to the basal cortex in anaphase (*Figure 4H and H'*). An antibody penetration issue with our Numb antibody in this set of experiments prevented us from performing a more in-depth analysis of cortical/cytoplasmic Numb signal ratios. In conclusion, phosphorylation of Baz-S180 is not necessary for the coordination of Baz localisation throughout the cell cycle but is involved in the timely establishment of basal polarity in larval neuroblasts.

## Phosphorylation of Baz-S180 affects sensory organ formation

As Baz-S180 is also phosphorylated in mitotic SOPs (*Figure 3D*), we next analysed the functional relevance of this Baz phosphorylation in these cells. During asymmetric cell division of SOPs, cortical polarity controls the asymmetric segregation of Notch regulators, causing differential activation of the Notch pathway and thus the acquisition of different cell identities. The individual RNAi-mediated depletion of Baz or the Notch ligand Delta do not result in severe sensory organ formation defects. However, co-depletion of both results in a neurogenic phenotype and the near complete loss of sensory organs, indicative of a Notch loss of function (*Houssin et al., 2021*). We reproduced this observation by depleting both *delta* and endogenous *baz::mScarlet-I* (*Figure 5A*), which enabled func-tional analysis of GFP-tagged Baz-S180 phosphomutants in this context. Expression of our Baz::GFP constructs partially rescued the formation of macrochaetes, large sensory organs formed at the begin-ning of pupal development, but surprisingly not that of microchaetes, smaller sensory organs formed later during pupal development (*Figure 5B*). The Pannier-GAL4 driver is expressed in a wide stripe running across the entire length of the notum and scutellum, in which UAS-driven RNAi to knock down Baz::mScarlet appeared to be efficient, as mScarlet signal appeared to be entirely depleted in this zone. Unexpectedly, the expression pattern of our UASz-driven Baz::GFP constructs was largely restricted to two narrow posterior stripes, potentially explaining their inability to rescue the formation of more anterior microchaetes (*Figure 5C*). As microchaetes did not form in this context, we focused our analysis on macrochaetes.

We observed that significantly more macrochaete bristles formed when expressing either phos-phomutant compared to the rescue with Baz^WT::GFP (*Figure 5D and D'*). This excess of bristles could be indicative of defects in lateral inhibition, a Notch-dependent mechanism restricting the number of sensory organ precursors. It could also result from defects in asymmetric cell divisions in the sensory organ precursor lineage leading to cell fate transformations such as socket-to-shaft transformations. To investigate both possibilities, we stained pupal nota for SOP lineage (Cut) and socket cell (Su(H)) markers (*Figure 5E*). Rescue by the Baz-S180 phosphomutants in the *baz, delta* double RNAi back-ground resulted in significantly more SOPs to be formed (*Figure 5F*) and in significantly more socket-to-shaft transformations (*Figure 5G*) than rescue by Baz^WT::GFP, suggesting a Notch loss of function

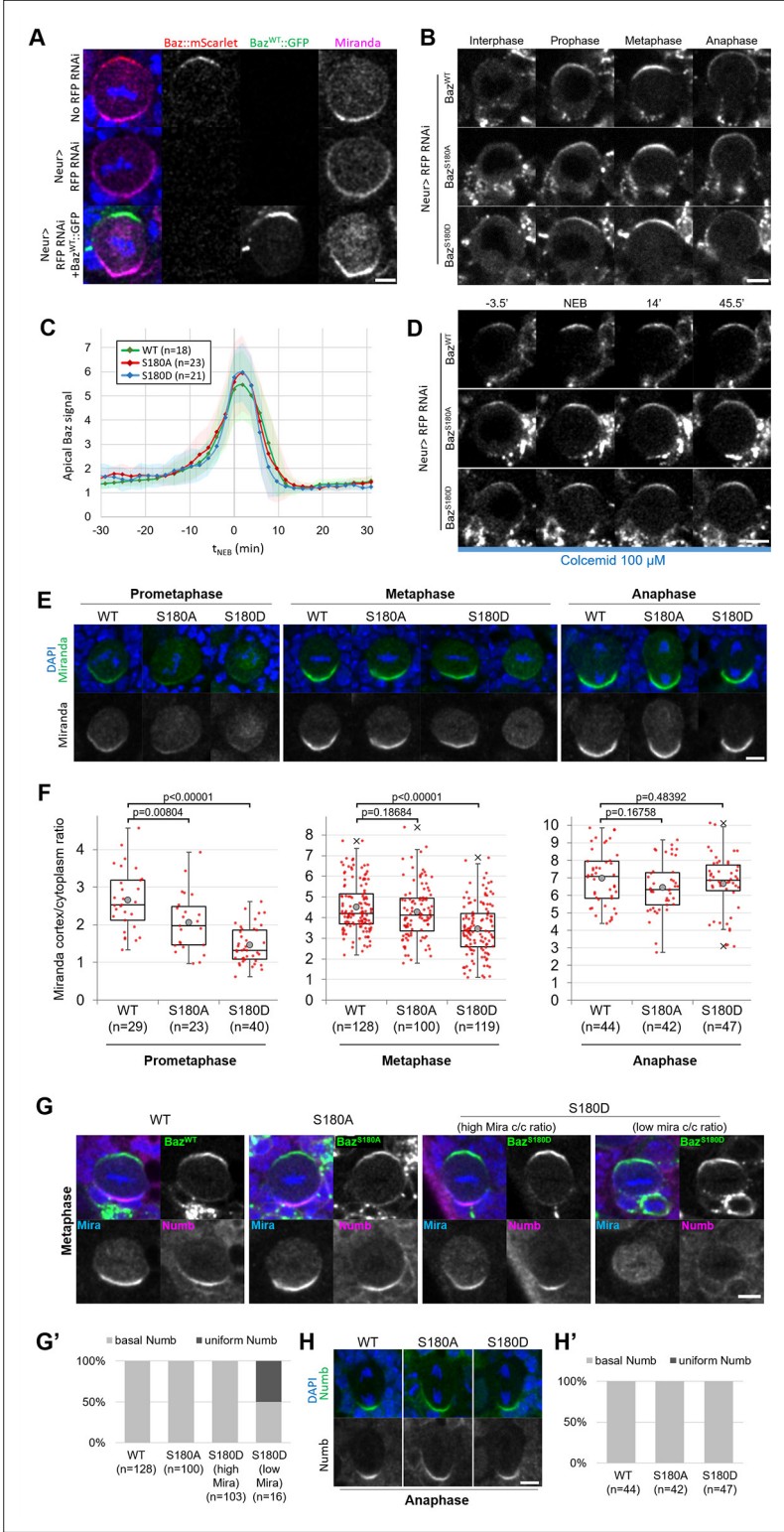

**Figure 4.** Localisation and function of Baz-S180 phosphomutants in neuroblasts. (**A**) Fixed metaphase neuroblasts expressing Baz::mScarlet (red) and stained for Miranda (magenta), with or without Neur-GAL4 driving RFP RNAi and Baz[WT]::GFP (green) expression. Scale bar: 5 µm. (**B**) Live neuroblasts depleted of endogenous Baz::mScarlet and expressing Baz[WT]::GFP, Baz[S180A]::GFP or Baz[S180D]::GFP. Scale bar: 5 µm. (**C**) Intensity of the apical Baz::GFP signal in cycling neuroblasts depleted of endogenous Baz::mScarlet, normalized to the intensity of the cytoplasmic Baz signal in interphase. t0: NEB. n=18 divisions (Baz[WT]::GFP), 23 divisions (Baz[S180A]::GFP) and

*Figure 4 continued on next page*

*Figure 4 continued*

21 divisions (Baz$^{S180D}$::GFP). 5 brains for all conditions, 2 experiments. Error bars: standard deviation. (**D**) Live metaphase-arrested neuroblasts depleted of endogenous Baz::mScarlet and expressing Baz$^{WT}$::GFP, Baz$^{S180A}$::GFP or Baz$^{S180D}$::GFP. T0: NEB. Scale bar: 5 μm. (**E**) Fixed neuroblasts depleted of endogenous Baz::mScarlet and expressing Baz$^{WT}$::GFP, Baz$^{S180A}$::GFP or Baz$^{S180D}$::GFP, stained for Miranda (green) and DAPI (blue). Two cases are displayed for Baz$^{S180D}$::GFP in metaphase, one showing polarized cortical Miranda (left) and the other mostly cytoplasmic Miranda (right). Scale bar: 5 μm. (**F**) Ratio between the basal Miranda cortical signal and the cytoplasmic signal. n=29 prophases, 128 metaphases, 44 anaphases in 17 brains (Baz$^{WT}$::GFP), 23 prophases, 100 metaphases, 42 anaphases in 18 brains (Baz$^{S180A}$::GFP), and 40 prophases, 119 metaphases, 47 anaphases in 18 brains (Baz$^{S180D}$::GFP). Statistical test: two-tailed Mann–Whitney U test. Box plots: cross: maximal and/or minimal outliers (beyond 1.5×interquartile range); grey circle: average; red dots: individual measurements; centre line, median; box limits, upper and lower quartiles; whiskers, 1.5×interquartile range. 2 experiments. (**G**) Fixed neuroblasts in metaphase, depleted of endogenous Baz::mScarlet and expressing Baz$^{WT}$::GFP, Baz$^{S180A}$::GFP or Baz$^{S180D}$::GFP (green), stained for Miranda (blue), Numb (magenta) and DAPI (blue). Two cases are displayed for Baz$^{S180D}$::GFP in metaphase, one showing polarized cortical Miranda (high Mira cortical/cytoplasmic ratio, left) and the other mostly cytoplasmic Miranda (low Mira c/c ratio, right). Scale bar: 5 μm. (**G′**) Proportions of cases with basal or uniform localisation of Numb in metaphase. (**H**) Fixed neuroblasts in metaphase, depleted of endogenous Baz::mScarlet and expressing Baz$^{WT}$::GFP, Baz$^{S180A}$::GFP or Baz$^{S180D}$::GFP, stained for Numb (green) and DAPI (blue). Scale bar: 5 μm. (**H′**) Proportions of cases with basal or uniform localisation of Numb in anaphase.

during lateral inhibition and following asymmetric cell divisions, respectively. Thus, in the absence of Delta, phosphorylation of Baz-S180 affects sensory organ formation both during the initial specification of SOPs and during the asymmetric cell divisions they undergo later.

The latter is consistent with the observation that Baz-S180 is phosphorylated in mitotic SOPs (*Figure 3D*). In contrast, we were puzzled by the involvement of Baz-S180 phosphorylation in SOP specification: the lateral inhibition process does not involve ACD, whereas Baz-S180 phosphorylation is restricted to ACD in the larval brain (*Figure 3C*) and in the pupal notum at the developmental stage when SOPs divide (*Figure 3D*). This prompted us to test whether Baz-S180 is phosphorylated earlier in the notum, during lateral inhibition. Microchaete SOPs are specified within Delta-positive stripes of proneural cells between 10 and 12 hr APF (*Usui and Kimura, 1992*; *Parks et al., 1997*). We observed patches of Baz-pS180-positive cells that overlapped with these Delta-positive stripes at 8 hr APF (*Figure 5H*) These observations are consistent with a role of Baz-S180 phosphorylation in lateral inhibition and indicate that it is not ACD-specific in this tissue at this developmental stage. Importantly, a strong Baz-pS180 signal was also present outside of these stripes, in particular close to the scutellar macrochaetes (*Figure 5H*, arrowhead), in a region devoid of microchaete SOPs. This suggests that Baz-S180 phosphorylation may have additional roles during development.

## Phosphorylation of Baz-S180 controls Baz localisation during pIIa division

To test whether Baz-S180 phosphorylation influences its localisation in SOPs and their lineage (*Figure 6A*), we used the Neur-GAL4 driver to express our Baz::GFP constructs and to deplete *delta* and endogenous *baz::mScarlet* by RNAi. Baz::mScarlet appeared to be efficiently depleted in microchaete SOPs in the pupal notum (*Figure 6B*) but adult flies did not have any sensory organ formation defect (not shown), perhaps because of an incomplete Delta depletion, which we did not assess. Surprisingly, we observed that only a subset of microchaete SOPs in the pupal notum expressed our Neur-GAL4, UASz-controlled Baz::GFP constructs (*Figure 6C*, *Figure 6—figure supplement 1A*). As SOPs and their progeny underwent asymmetric cell divisions, Baz::GFP started to be gradually expressed in other SOPs (*Figure 6C*), but, in most cases, not in their entire lineage. Baz::GFP sometimes started after the pI division in both its daughter cells, but was more often restricted to the pIIb (*Figure 6—figure supplement 1B*), pIIa (*Figure 6—figure supplement 1C*) or pIIIb lineage (*Figure 6—figure supplement 1D*). The Baz-S180 phosphomutants localisation did not appear to be affected during the ACD of pI: in metaphase, they formed posterior crescents which intensity was not significantly different from Baz$^{WT}$::GFP (*Figure 6D and D′*). The unexpected spatio-temporal mosaicism of our Baz::GFP transgenes within the SOP lineage gave us the opportunity to measure cortical Baz::GFP levels during pIIa division without any contribution from its sister cell pIIb, in cases where Baz::GFP expression was limited to the pIIa lineage (*Figure 6E*). In contrast to pI division, the Baz-S180

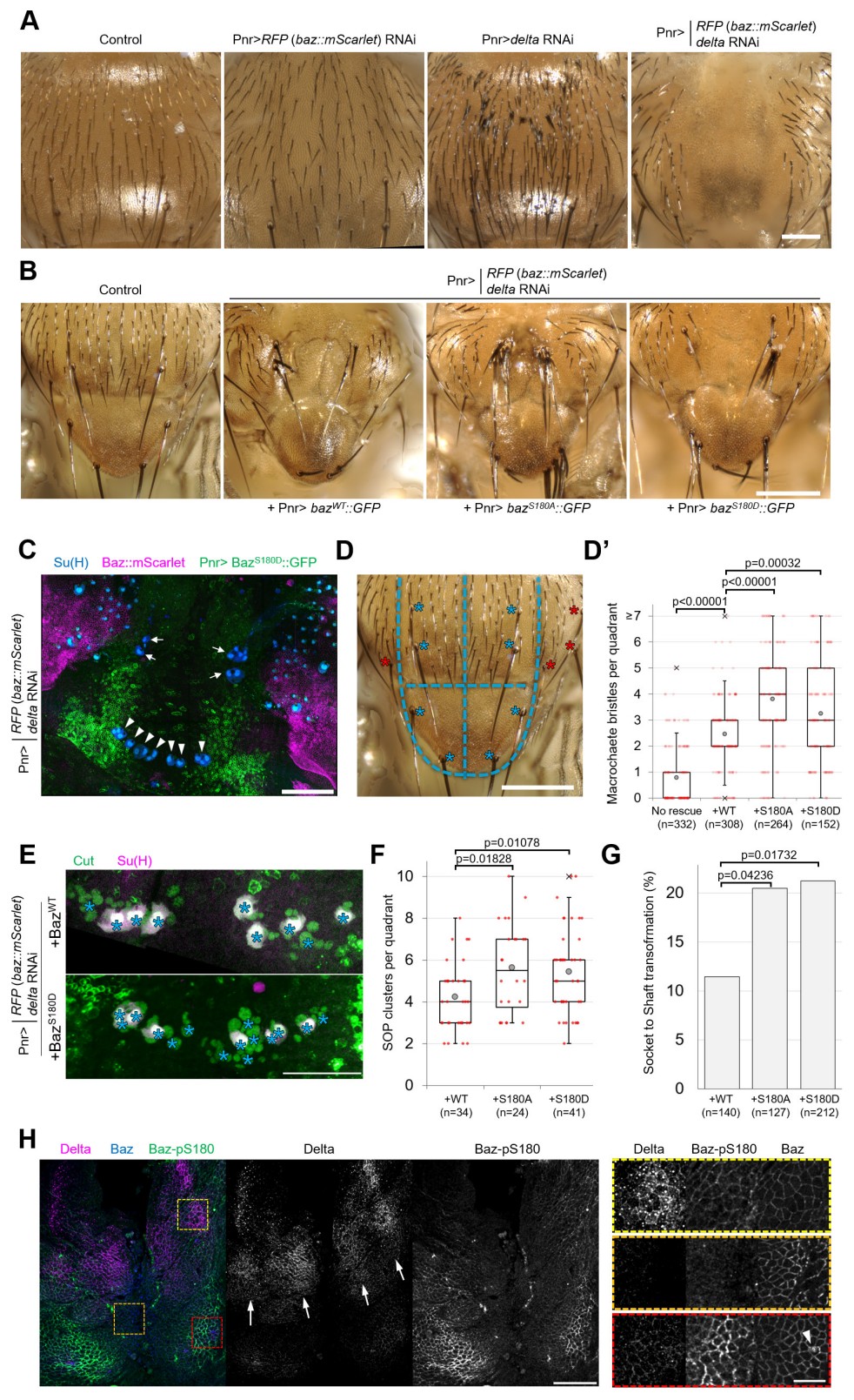

**Figure 5.** Baz-S180 phosphorylation regulates sensory organ formation. (**A**) Adult nota with or without Pannier-GAL-driven RNAi of *baz::mScarlet* and/or *delta*. Scale bar: 100 μm. (**B**) Control adult notum and *baz::mScarlet, delta*-depleted nota expressing Baz^WT^::GFP, Baz^S180A^::GFP or Baz^S180D^::GFP. Scale bar: 100 μm. (**C**) Pupal notum stained for Su(H) (blue). Pannier-GAL4 drives both the depletion of endogenously expressed Baz::mScarlet and

*Figure 5 continued on next page*

*Figure 5 continued*

the expression of Baz$^{S180D}$::GFP. Arrows: dorso-central macrochaetes. Arrowheads: scutellar macrochaetes. Scale bar: 100 µm. (**D**) Quadrants defined for macrochaete bristles counting within the Pannier-GAL4 expression domain. The two upper quadrants include dorso-central macrochaetes and the two lower quadrants include scutellar macrochaetes (blues asterisks). Other macrochaetes (red asterisks) outside of the Pannier-GAL4 expression domain were ignored. Scale bar: 100 µm. (**D'**) Number of individual bristles per quadrant in *baz::mScarlet, delta*-depleted adult nota, with or without expression of Baz::GFP transgenes. Statistical test: two-tailed Mann–Whitney U test. Box plot: cross: maximal and/or minimal outliers (beyond 1.5×interquartile range); grey circle: average; red dots: individual measurements; centre line, median; box limits, upper and lower quartiles; whiskers, 1.5×interquartile range. n=332 quadrants in 83 flies (no rescue), 308 quadrants in 77 flies (Baz$^{WT}$::GFP), 264 quadrants in 66 flies (Baz$^{S180A}$::GFP) and 152 quadrants in 38 flies (Baz$^{S180D}$::GFP). (**E**) Pupal notum stained for Cut (green) and Su(H) (magenta). Asterisks: individual SOP clusters. Scale bar: 50 µm. (**F**) Number of individual SOP clusters per quadrant in *baz::mScarlet, delta*-depleted pupal nota expressing Baz::GFP transgenes. Statistical test: two-tailed Mann–Whitney U test. Box plot: cross: maximal and/or minimal outliers (beyond 1.5×interquartile range); grey circle: average; red dots: individual measurements; centre line, median; box limits, upper and lower quartiles; whiskers, 1.5×interquartile range. N=34 quadrants in 9 nota (Baz$^{WT}$::GFP), 24 quadrants in 6 nota (Baz$^{S180A}$::GFP), and 41 quadrants in 11 nota (Baz$^{S180D}$::GFP). 4 experiments. (**G**) Percentage of cases of Socket cell to Shaft cell transformation cases in *baz::mScarlet, delta*-depleted pupal nota expressing Baz::GFP transgenes. n=140 SOP clusters in 9 nota (Baz$^{WT}$::GFP), 127 SOP clusters in 6 nota (Baz$^{S180A}$::GFP), and 212 SOP clusters in 11 nota (Baz$^{S180D}$::GFP). 4 experiments. Statistical test: two-tailed Z score calculation of population proportions. (**H**) 8 h APF notum expressing Delta::GFP (magenta) and Baz::mScarlet (blue) stained for Baz-pS180 (green). Right panels: close-ups of the boxed areas in left panels. Scale bars: 50 µm (left) and 10 µm (right). Arrows: Delta-positive stripes. Arrowhead: scutellar macrochaete.

phosphomutants localisation differed from Baz$^{WT}$::GFP during pIIa metaphase. Like Baz$^{WT}$::GFP, Baz$^{S180A}$::GFP formed posterior crescents, but with a significantly weaker cortical-to-cytoplasmic ratio. Baz$^{S180D}$::GFP crescents appeared enlarged and displaced toward the anterior pole, leading to a significant increase of the anterior cortical signal (***Figure 6E and E'***). Therefore, Baz localisation during the ACD of pIIa but not its mother cell pI is regulated by Baz-S180 phosphorylation.

In addition to apical junctions, Baz localises to two SOP-specific structures at the interface between pIIa and its sister cell pIIb (or, following pIIb division, at the pIIa/pIIIb interface): first, in basolateral nano-clusters (***Figure 6—figure supplement 1A***, arrows) involved in Notch signalling (***Houssin et al., 2021***) second, in a cortical patch forming during pIIa division (***Figure 6—figure supplement 1A***, arrowhead) and involved in the regulation of its division axis (***Le Borgne et al., 2002***). We took advantage of mosaic Baz::GFP expression to analyze the contribution of pIIa and pIIb (or pIIIb) to both structures. Baz-positive puncta were present at the basolateral pIIb/pIIa interface whether Baz::GFP was expressed only in pIIb (***Figure 6—figure supplement 1E***) or pIIa (***Figure 6—figure supplement 1F***). In contrast, Baz::GFP only localised to the cortical patch forming during pIIa division when it was expressed in pIIb (***Figure 6—figure supplement 1B–C***). Thus, pIIb and pIIa both contribute to basolateral Baz nanocluster (***Figure 6—figure supplement 1G***, arrows), whereas the Baz present at the cortical patch during pIIa division is exclusively present in pIIIb (***Figure 6—figure supplement 1G***, arrowhead). Finally, we noticed one functional consequence of mosaic Baz::GFP expression. Unlike the pI and pIIa cells which orient their divisions within the plane of the epithelium, pIIb divides perpendicularly to the plane of the epithelium (***Gho et al., 1999***). We observed that pIIb also tended to do so in most cases where Baz::GFP was present in both pIIb and pIIa. Surprisingly, when Baz::GFP was present in pIIb but not in pIIa, pIIb tended to divide within the plane of the epithelium (***Figure 6—figure supplement 1H–I***). In all the cases where the imaging session lasted long enough for this, we observed that one of the daughter cells (pIIIb) resulting from a planar pIIb division divided later on (n=4/4, ***Figure 6—figure supplement 1H***), indicating that planar pIIb divisions are not caused by a pIIb-to-pIIa transformation causing pIIb to orient its division in a pIIa-like manner, as pIIa daughter cells do not divide (***Figure 6A***). Thus, the orientation of the pIIb division is controlled in a cell non-autonomous manner by the levels of Baz in its sister cell pIIa.

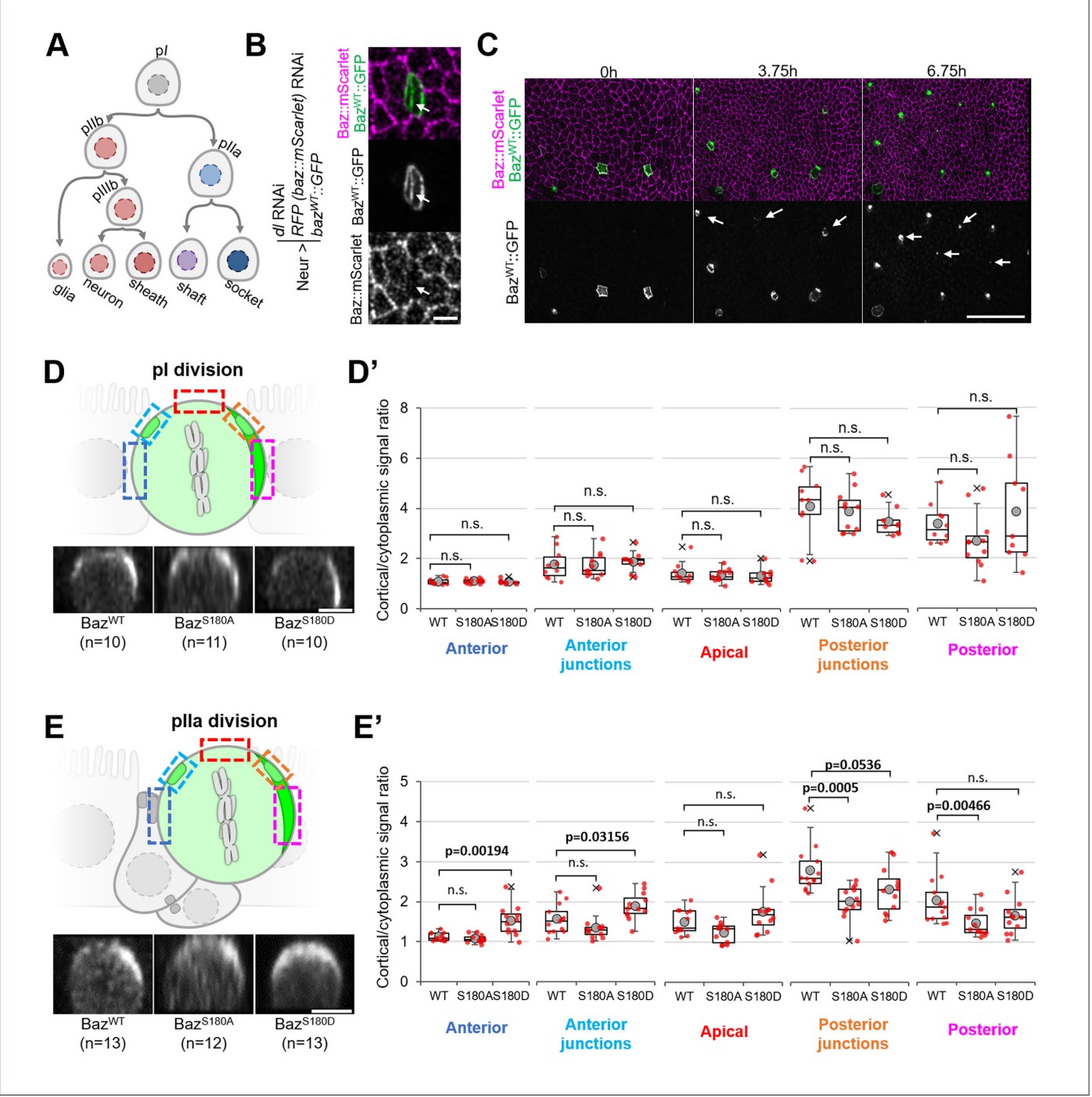

**Figure 6.** Baz-S180 phosphomutants localisation during SOP lineage divisions. In all panels, Neur-GAL4 drives *delta* and *baz::mScarlet* RNAi as well as Baz^WT/S180A/S180D^::GFP expression and live pupal nota are shown. For Baz^WT^: 4 nota analysed in 2 independent experiments. For Baz^S180A^: 3 nota in 2 experiments. For Baz^S180D^: 3 nota in 2 experiments. (**A**) SOP lineage. Diverging arrows: asymmetric cell divisions. (**B**) SOP cluster at the two-cell (pIIb/pIIa) stage. Arrow: Baz::mScarlet (magenta) is not detectable at the pIIb/pIIa interface. (**C**) Pupal notum during SOP lineage divisions. Arrows: SOP clusters starting to express Baz::GFP during the imaging session. Scale bar: 50 µm. (**D**) Orthogonal view of pI division at metaphase. Apical is up and anterior is left. Scale bar: 5 µm. Boxed areas: cortical areas measured. (**D'**) Cortical/cytoplasmic Baz^WT/S180A/S180D^::GFP signal during pI division at different locations of the cortex. Statistical test: two-tailed Mann–Whitney U test. Box plot: cross: maximal and/or minimal outliers (beyond 1.5×interquartile range); grey circle: average; red dots: individual measurements; centre line, median; box limits, upper and lower quartiles; whiskers, 1.5×interquartile range. (**E**) Orthogonal view of pIIa division at metaphase in cases where Baz^WT/S180A/S180D^::GFP is not expressed in the pIIb lineage. Apical is up and anterior is left. Scale bar: 5 µm. Boxed areas: cortical areas measured. (**E'**) Cortical/cytoplasmic Baz^WT/S180A/S180D^::GFP signal during pIIa division at different locations of

*Figure 6 continued on next page*

*Figure 6 continued*

the cortex. Statistical test: two-tailed Mann–Whitney U test. Box plot: cross: maximal and/or minimal outliers (beyond 1.5×interquartile range); grey circle: average; red dots: individual measurements; centre line, median; box limits, upper and lower quartiles; whiskers, 1.5×interquartile range.

The online version of this article includes the following figure supplement(s) for figure 6:

**Figure supplement 1.** In all panels, Neur-GAL4 drives *delta* RNAi, *baz::mScarlet* RNAi and Baz[WT/S180A/S180D]::GFP expression and live pupal nota are shown.

## *Drosophila* Baz-S180 and human PARD3-S187 are phosphorylated by CDK1/CyclinB1 in vitro

Given that Baz-S180 is a predicted consensus CDK phosphosite, we next investigated whether it can be phosphorylated by CDK1. We synthesised a Baz fragment N-terminal to the first PDZ domain (Baz[2-320], *Figure 7A*). We first used western blotting with our phospho-specific anti-Baz-pS180 antibody to test whether incubation of Baz[2-320] with CDK1/CyclinB1 complexes results in Baz-S180 phosphorylation. Indeed, although the phospho-specific Baz-PS180 slightly bound to Baz[2-320] in the absence of CDK1/CyclinB1 or ATP, incubation with CDK1/CyclinB1 resulted in a large increase of anti-Baz-pS180 binding (*Figure 7B*). We next used targeted mass spectrometry to further examine Baz[2-320] Ser/Thr phosphorylation (*Figure 7A*, *Supplementary file 1*). We conclude from these results that CDK1 phosphorylates Baz-S180 in vitro.

Interestingly, although the sequence of the disordered region surrounding S180 does not appear to be conserved in the Baz human ortholog PARD3, we identified on PARD3 a full consensus CDK phosphosite (S187) in close proximity, like S180 in Baz, to a 14-3-3 binding site (*Figure 7C*). PARD3-S187 was previously found phosphorylated in high-throughput mass spectrometry studies (*Lin et al., 2021*). This prompted us to test whether PARD3-S187 is also phosphorylated by CDK1. As we did for Baz, we synthesised a PARD3 fragment N-terminal to the first PDZ domain (PARD3[2-281], *Figure 7A*). Using mass spectrometry, we observed that PARD3-S187 was only phosphorylated in the presence of CDK1/CyclinB1 complexes, along with a few other residues not matching consensus CDK phosphosites (*Figure 7A*, *Supplementary file 1*). Thus, CDK1 phosphorylates human PARD3 in vitro on a Serine which localisation closely matches Baz-S180.

## Discussion

### Partial inhibition of CDK1 does not abolish neuroblast polarity

Here, we first reinvestigated the role of CDK1 during neuroblasts asymmetric cell division. Based on the observation on fixed tissues that dominant negative or thermosensitive *cdk1* alleles do not prevent embryonic neuroblasts from cycling but disrupt their polarity, asymmetric cell division was proposed to strictly rely on high levels of CDK1 activity (*Tio et al., 2001*). However, using live imaging analysis and chemical genetics, we did not reproduce these observations: the apical polarity protein Baz still polarizes upon partial inhibition of CDK1 using an analog-sensitive allele of *cdk1*, both in embryonic and larval neuroblasts (*Figure 2*). A possible cause for this discrepancy could be that loss of Baz polarity was previously only reported with the dominant negative *cdk1[E51Q]* allele, for which neomorphic activity cannot be ruled out, whereas loss of polarity in the thermosensitive *cdk1* genetic situation was only reported for another apical polarity protein, Inscuteable. Considering that Baz functions upstream of Inscuteable (*Schober et al., 1999*), it is possible that Baz localisation is not affected in this situation.

### High CDK1 activity triggers apical coalescence

While partial inhibition using the analog-sensitive allele of cdk1 did not abolish Baz polarity, it did prevent apical coalescence: large apical crescents formed during prophase failed to coalesce into smaller crescents at NEB (*Figure 2*). Although our observations show that high CDK1 activity is required for apical coalescence, the exact mechanism regulating this process remains elusive. Our previous observation that aPKC inhibition can result in apical crescents coalescing in the wrong direction (*Hannaford et al., 2019*) suggests that apical polarity proteins are involved in some aspects of this process. As coalescence was not affected in Baz-S180 phosphomutants (*Figure 4*), CDK1-dependent control of apical coalescence might be controlled by other CDK1 phosphosites on Baz

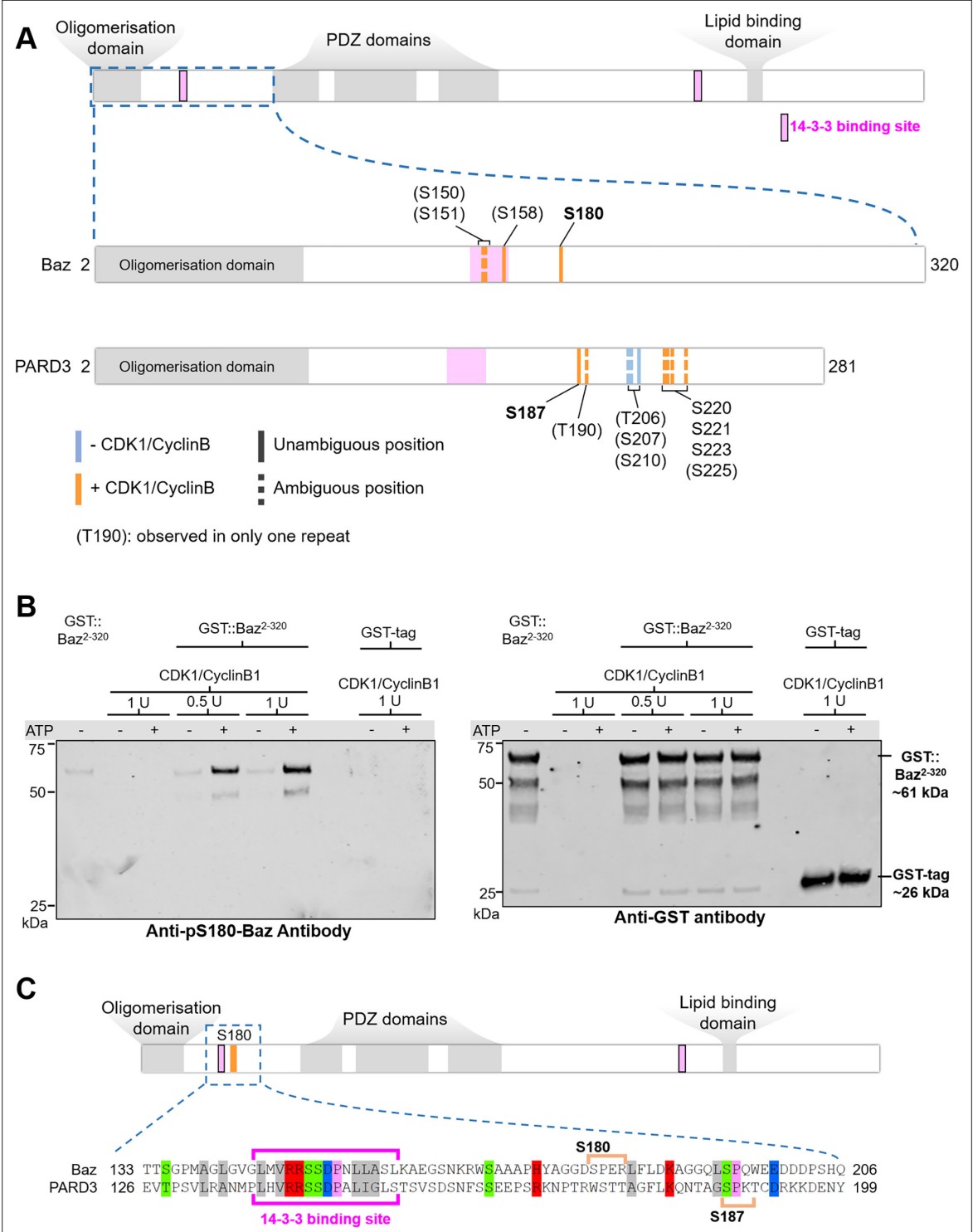

**Figure 7.** *Drosophila* Baz-S180 and human PARD3-S187 are substrates of CDK1 in vitro. (**A**) Domains of Baz/PARD3 (up) and close-ups (down) of the N-Terminal fragments used to test phosphorylation of Baz by CDK1 in vitro by mass spectrometry. Blue bars: residues found phosphorylated in the absence or presence of CDK1/CyclinB. Orange bars: residues found phosphorylated only in the presence of CDK1/CyclinB. Solid bars: unambiguous position. Dashed bars: ambiguous position. Positions between parenthesis: residues found phosphorylated in only one of the two repeats. (**B**) Western blot showing phosphorylation of a purified GST-Baz²⁻³²⁰ fragment on Serine180 following incubation with human CDK1/CyclinB. Left panel: Baz-pS180 antibody (*Supplementary file 1*). Right panel: reprobing with an anti-GST antibody. (**C**) Alignment of the Baz sequence around S180 with human PARD3. Orange brackets: consensus CDK phosphosites. The western blots were repeated once yielding the same result.

*Figure 7 continued on next page*

*Figure 7 continued*

The online version of this article includes the following source data for figure 7:

**Source data 1.** CyclinB1/CDK1 phosphorylates a Baz fragment on S180 in vitro.

or other CDK1 targets. RhoGAPs and RhoGEFs are promising candidates for this: first, apical coalescence of neuroblasts is driven by basal-to-apical actomyosin-driven cortical flows (*Oon and Prehoda, 2019*), reminiscent of the ones polarising *C. elegans* embryos (*Munro et al., 2004*) induced by asymmetric localisation of the RhoGEF ECT-2 (*Motegi and Sugimoto, 2006*) second, ECT-2 can be phosphorylated by CDK1 (*Niiya et al., 2005*). Based on the observation that some RhoGEFs such as the fly ECT-2 homolog Pebble can be sequestered in the nucleus until NEB and that apical coalescence occurs at NEB, a tempting model for the temporal regulation of coalescence would be that the release of RhoGEFs at NEB triggers a basal-to-apical flow. However, we showed that NEB and apical coalescence can be temporally uncoupled by first partially inhibiting CDK1, then restoring its full activity after NEB has occurred (*Figure 2F*). Thus, although an involvement of NEB cannot be ruled out in its temporal control, it is ultimately high CDK1 activity that triggers apical coalescence. Interestingly, enlarged Baz crescents were observed in *aurora A* mutants (*Wang et al., 2006*). Given that CDK1 activates Aurora A (*Van Horn et al., 2010*), this raises the possibility that CDK1 drives apical coalescence via Aurora A activation.

### Function of apical coalescence

Beyond its regulation, the function of apical coalescence is unknown. Based on our observation that partial inhibition of CDK1 not only leads to defective coalescence but also to crescent instability in metaphase in cycling neuroblasts (*Figure 2D*), we initially speculated that coalescence, by concentrating apical polarity proteins, might reinforce positive feedback loops for apical protein concentration and thus stabilize them. However, partial inhibition of CDK1 and the resulting failure to coalesce surprisingly did not seem to affect crescent stability in metaphase-arrested neuroblasts (*Figure 2E*). A possible explanation for this difference between cycling and arrested cells could be that, in cycling cells, partial inhibition of CDK1 somehow prematurely triggers some anaphase events involving Baz crescents disassembly, uncoupling them from other anaphase events such as DNA segregation and cortical furrowing. Regardless of the reason for this difference, apical coalescence is not necessary for apical polarity maintenance. However, we did observe that basal Numb crescents significantly increased in intensity upon partial inhibition of CDK1 (*Figure 2H and K*). We speculate that enlarged apical crescents might be able to exclude basal protein further down the apico-basal axis, concentrating basal components in smaller basal crescents. Brighter Numb crescents did not appear noticeably smaller than in controls in our fixed tissue analysis, but live imaging of neuroblasts in culture would be necessary to accurately measure basal crescents size. An alternative explanation could be that, upon partial inhibition of CDK1, a longer cell cycle allows neuroblasts to synthesize more Numb, raising the interesting possibility of a modulation of the strength of basal asymmetry by cell cycle duration. Finding ways of interfering with coalescence without increasing cell cycle length would be necessary to test this. Finally, it is possible that loss of basal-to-apical flows could affect the properties of the basal cortex, impacting basal fate determinants localisation.

### Phosphorylation of Baz-S180 by CDK1

We confirmed in vitro that Baz-S180 is phosphorylated by CDK1 (*Figure 7A and B*), as suggested by its consensus CDK phosphorylation motif near Cyclin and Cks1 binding sites (*Figure 3A*). This seems consistent with our observation that Baz-S180 is phosphorylated in asymmetrically dividing neuroblasts and SOPs. However, CDK1 is also involved during symmetric cell divisions, in which we could not detect any Baz-S180 phosphorylation (*Figure 3B and C*, *Figure 3—figure supplement 1*). This difference might result from higher CDK1 activity than in symmetrically dividing cells, from an asymmetric cell division-specific adaptor between CDK1 and Baz, or from asymmetric celldivision-specific inhibition of Baz-S180 dephosphorylation. It is also noteworthy that Cyclin A colocalises with Baz in asymmetrically dividing SOPs (*Darnat et al., 2022*, *Figure 3—figure supplement 1*), perhaps explaining SOP-specific Baz-S180 phosphorylation. However, none of the CDK1-associated Cyclins localise asymmetrically to the mitotic neuroblasts cortex (*Figure 3—figure supplement 1*).

Alternatively, Baz-S180 could simply be phosphorylated by another as-yet-unknown kinase specifically expressed in asymmetrically dividing cell and only active during mitosis. Despite only being observed during ACD in larval brains and the pupal notum at 16 hr APF (*Figure 3*), Baz-S180 phosphorylation is not strictly ACD-specific throughout development, as we observed it in large patches of non-mitotic epithelial cells in the early pupal notum at 8 hr APF (*Figure 5H*). Although these cells are not mitotic, we cannot exclude that this phosphorylation is mediated by CDK1: all cells of the notum are arrested in G2 at this developmental stage (*Hunter et al., 2016*) and CDK1 can be active in G2 (*Hochegger et al., 2008*). Again, this does not explain why Baz-S180 is not phosphorylated in all these G2-arrested cells (*Figure 5H*), and it is possible that another kinase is responsible for the Baz-S180 phosphorylation pattern of 8 hr APF nota.

Importantly, we found that CDK1 phosphorylates in vitro Serine S187 of PARD3, the human ortholog of Baz (*Figure 7A*). PARD3-S187 is, like Baz-S180 in close proximity to an N-terminal 14-3-3 binding site (*Figure 7C*) regulating PAR-3 proteins localisation (*Benton and St Johnston, 2003*). Whether PARD3-S187 is also phosphorylated in vivo in a tissue-specific, cell cycle stage-specific and/ or ACD-specific manner in humans remains to be determined.

## Function of Baz-S180 phosphorylation in neuroblasts

We used Baz-S180 phosphomutants in the absence of endogenous Baz to investigate the function of this phosphorylation, expecting it to be necessary for the timing of Baz polarization based on its mitosis-specificity, and/or for Baz ability to polarize, given its asymmetric cell division-specificity. However, neither polarization itself, nor its timing, nor its maintenance were affected in Baz-S180 phosphomutant neuroblasts (*Figure 4B–D*). Thus, Baz-S180 phosphorylation is not the temporal cue coordinating cell polarity and the cell cycle (or at least not the only one), and the observed effects upon full (*Figure 1*) or partial (*Figure 2*) CDK1 inhibition on Baz localisation and dynamics are likely caused by other phosphosites on Baz and/or other CDK1 substrates. Although apical Baz crescents were apparently unaffected in Baz-S180 phosphomutants, the formation of basal Mira and Numb crescents was affected in metaphase. In contrast, basal crescents were indistinguishable from controls in anaphase (*Figure 4E–H'*), perhaps through the 'telophase rescue' mechanism that somehow restores defective basal polarity in *baz* and *insc* mutant neuroblasts (*Schober et al., 1999*; *Wodarz et al., 1999*).

Why the establishment of basal polarity is delayed in Baz-S180 phosphomutants remains unclear. As the function of the PAR complex is to exclude Miranda from the apical pole by phosphorylating it, defective recruitment of Miranda to the cortex (*Figure 4E*) suggests a gain of function of the PAR complex in Baz-S180 phosphomutants. We speculate that it might be caused by an excessive transfer of aPKC from an immobile apical Baz/aPKC/PAR-6 complex to a more diffusible Cdc42/aPKC/PAR-6 complex (*Rodriguez et al., 2017*), able to phosphorylate Miranda further down the apico-basal axis. In contrast, uniform localisation of Numb in metaphase (*Figure 4G*) suggests a loss of function of the PAR complex. This stark difference between Numb and Mira might come from the fact that, unlike Mira, Numb localisation is not only controlled by phosphorylation by the PAR complex (*Smith et al., 2007*; *Wirtz-Peitz et al., 2008*), but also by Polo-mediated phosphorylation of its binding partner Pon (*Wang et al., 2007*).

## Function of Baz-S180 phosphorylation in sensory organs formation

We showed that sensory organs formation is affected in Baz-S180 phosphomutants: we observed an excess of bristles (*Figure 5D'*) caused by the specification of too many sensory organ precursor cells (*Figure 5F*) and by lineage transformations leading to an excess of shaft cells (*Figure 5G*). These phenotypes are consistent with Baz-S180 being phosphorylated in the notum during both SOP specification at 8 hr APF (*Figure 5H*) and asymmetric SOP divisions at 16 hr APF (*Figure 3D*). Both phenotypes are classically caused by a Notch loss of function. When SOPs are specified through Notch-dependent lateral inhibition, failure to activate Notch in their neighbouring cells allows them to also be specified as SOPs, leading to supernumerary sensory organs (*Simpson, 1990*). Later on, failure to activate the Notch pathway following ACDs taking place in the SOP lineage leads to cell fate transformations (*Schweisguth, 2015*), such as socket-to-shaft transformations causing one sensory organ to produce two bristles. Notch loss of function during ACD in the Baz phosphomutants could be due to defective cell polarity-controlled asymmetric segregation of Notch regulators (*Schweisguth, 2015*), or to

defective Baz-dependent recruitment of Notch at the interface of daughter cells following ACD (*Wu et al., 2023b*; *Houssin et al., 2021*). We speculate that Baz involvement in lateral inhibition may be mediated by a similar, yet untested Baz-dependent recruitment of Notch or its ligands at the interface between Notch signal-sending and signal-receiving cells during this process. Baz was also shown to be somehow involved in the activation of the Notch intracellular domain following its cleavage (*Wu et al., 2023a*), which may have implications for the regulation of both lateral inhibition and ACD.

Unlike in neuroblasts, Baz-S180 phosphorylation controls its localisation in SOPs, at least during ACD of the pIIa cell. During pIIa metaphase, Baz$^{S180A}$ crescents were weaker than controls and Baz$^{S180D}$ crescents were displaced (*Figure 6E*). In contrast, Baz phosphomutants crescents appeared normal during pI division (*Figure 6D*). This difference may come from different mechanisms controlling division orientation *via* cell polarity: pI polarization is oriented at the tissular level by planar cell polarity (*Gho and Schweisguth, 1998*), whereas pIIa polarization is oriented by an E-Cadherin-rich cortical patch allowing it to retain the division orientation of its mother cell pI (*Le Borgne et al., 2002*). Our results suggest that Baz-S180 phosphorylation is specifically involved in this division axis maintenance mechanism in pIIa. It is noteworthy that, with the exceptions of basal polarization being more delayed in *baz$^{S180D}$* than in *baz$^{S180A}$* in neuroblasts as well as Baz polarization being differently affected in *baz$^{S180A}$* and *baz$^{S180D}$* during pIIa division in SOPs, the Baz-S180 non-phosphorylatable and phosphomimetic mutants give the same phenotypes. This is likely because phosphomimetics do not always faithfully reproduce the effects of a phosphorylation (*Gogl et al., 2021*).

## Robustness of Baz localisation and function

Across the few decades during which Baz has been studied, a recurring observation is that Baz-dependent processes are often robust, involving context-specific functional redundancies between Baz and other proteins. Therefore, despite the central importance of Baz throughout development, disrupting Baz function alone can sometimes lead to mild phenotypes which are significantly worsened by concomitantly interfering with other proteins. Accordingly, Baz functions redundantly with the polarity protein PAR-6 in tracheal terminal cells branching (*Jones and Metzstein, 2011*), with the transmembrane protein Crumbs in the maintenance of epithelial polarity (*Müller and Wieschaus, 1996*), with the Notch ligand Delta in the formation of sensory organs (*Houssin et al., 2021*), and with the apical polarity Pins in the generation of size asymmetry during neuroblasts asymmetric cell division (*Cai et al., 2003*). Beyond redundancies with other proteins, the robustness of Baz function relies on a lipid binding domain and an oligomerization domain, which are both sufficient on their own to support Baz localisation and function (*Kullmann and Krahn, 2018*). In this light, it is perhaps not surprising that the phenotypes that we describe in neuroblasts remain relatively mild: partial inhibition of CDK1 does not abolish apical nor basal polarity (*Figure 2*), and Baz-S180 phosphomutants display no localisation defect and only a delay in basal polarity establishment (*Figure 3*). It is conceivable that, in these situations, Baz localisation and/or function is rescued by another protein. A very likely candidate for this is Pins, known to function redundantly with Baz in neuroblasts (*Cai et al., 2003*; *Yu et al., 2000*; *Izumi et al., 2004*). Could Pins be phosphorylated by mitotic kinases, and in this case, would both Baz and Pins need to lose the temporal cue provided by phosphorylation to uncouple cell polarity from the cell cycle? In the absence of Pins, would apical coalescence become necessary for the maintenance of apical polarity? Would Baz-S180 phosphomutants fail to polarize? Addressing these questions will be the focus of further studies.

## Materials and methods

### Key resources table

| Reagent type (species) or resource | Designation | Source or reference | Identifiers | Additional information |
|---|---|---|---|---|
| Gene (*Drosophila melanogaster*) | mira::mCherry | CrispR edited locus by Januschke lab; *Ramat et al., 2017* | | |
| Gene (*Drosophila melanogaster*) | baz::mScarlet-I | CrispR edited locus by Januschke lab; *Houssin et al., 2021* | | |

*Continued on next page*

*Continued*

| Reagent type (species) or resource | Designation | Source or reference | Identifiers | Additional information |
|---|---|---|---|---|
| Gene (*D. melanogaster*) | cdk1as2 | CrispR edited locus by Januschke lab | This study | |
| Gene (*D. melanogaster*) | Baz::GFP | Bloomginton Stock Center | BDSC 51572 | |
| Gene (*D. melanogaster*) | ubi-bazΔLB::GFP | *Kullmann and Krahn, 2018* | | |
| Gene (*D. melanogaster*) | ubi-bazΔOD::GFP | *Kullmann and Krahn, 2018* | | |
| Gene (*D. melanogaster*) | ubi-his2av::mRFP | Gonzalez lab | Gift from Cayetano Gonzalez | |
| Gene (*D. melanogaster*) | UASz-bazWT::GFP | | This study | |
| Gene (*D. melanogaster*) | UASz-bazS180A::GFP | | This study | |
| Gene (*D. melanogaster*) | UASz-bazS180D::GFP | | This study | |
| Gene (*D. melanogaster*) | Neur-GAL4 | Bloomington stock center | BDSC 80575 | |
| Gene (*D. melanogaster*) | Pnr-GAL4 | Bloomington stock center | BDSC 3039 | |
| Gene (*D. melanogaster*) | UAS-RFP RNAi | Bloomington stock center | BDSC 67852 | |
| Gene (*D. melanogaster*) | UAS-dl RNAi | Bloomington stock center | BDSC 28032 | |
| Gene (*D. melanogaster*) | cycB::GFP | Bloomington stock center | BDSC 51568 | |
| Gene (*D. melanogaster*) | cycB3::GFP | Bloomington stock center | BDSC 91673 | |
| Gene (*D. melanogaster*) | Neur-GAL4, UAS-Pon::RFP | *Emery et al., 2005* | | |
| Gene (*D. melanogaster*) | delta::GFP | Bloomington stock center | BDSC 59819 | |
| Chemical compound, drug | 1-NAPP-1 | Sigma Aldrich | 529579 | |
| Antibody | affinity purified Baz-PS180 antibody | sheep | This study | YAGGDS*PERLF – (where S* is phospho-Serine) |
| Antibody | Affinity purified Numb antibody | sheep | This study | GST-full length Numb protein (CG3779-PA) used as antigen |
| Antibody | Affinity purified Mira antibody | sheep | This study | RLFRTPSLPQRLR peptide used as antigen |
| Antibody | Affinity purified Mira antibody | rabbit; *Ramat et al., 2017* | | |
| Antibody | rabbit anti-PKC ζ | Santa Cruz | sc-17781 | |
| Antibody | Mouse anti-Cyclin A | DSHB | DSHB A12 | |
| Antibody | Mouse anti-Su(H) | Santa Cruz | sc-398453 AF647 | |
| Antibody | Mouse anti-Cut | DSHB | DSHB 2B10 | |
| Recombinant DNA reagent | UASz 1.0 | DGRC | DGRC 1431 | |

## Materials availability statement

All reagents created in this study are made available upon request to the corresponding author.

## Fly stocks and genetics

Flies were reared on standard corn meal food at 25 °C, except for RNAi-expressing larvae and their corresponding controls (*Figures 4 and 5*, *Figure 3—figure supplement 1*), which were placed at 29 °C from the L1 larval stage to the L3 larval or pupal stage, at which point they were dissected, or the adult stage, at which point they were observed under anesthesia with $CO_2$. For the genotypes of the *Drosophila* lines used in each experiment, see *Supplementary file 2* For the origins of the stocks used, see *Supplementary file 3*.

## Immunostainings

The Baz-PS180 antibody was raised in sheep that were injected with YAGGDS*PERLF – (where S* is phospho-Serine). The fourth bleed was affinity purified using negative selection against a

YAGGDSPERLF peptide and positive selection using the YAGGDS*PERLF phospho-peptide. The poly-clonal Numb antibody was generated in sheep using GST-full length Numb protein (CG3779-PA) and affinity purified. Specificity was tested by staining mitotic *numb*[15] (*Le Borgne and Schweisguth, 2003*) mutant SOP clones in the notum, where immune reactivity dropped to background levels. Polyclonal anti Mira antibody was raised in rabbit using the CSPPQKQVLKARNI peptide and affinity purified. Specificity was tested in mitotic *mira*[KO] (*Ramat et al., 2017*), larval neuroblast clones where immune reactivity dropped to background levels. Polyclonal anti Mira antibody raised in sheep was generated using a RLFRTPSLPQRLR peptide followed by affinity purification.

For all immunostainings except for Baz-PS180 phosphostainings, larval brains, larval imaginal disks or pupal nota were dissected in phosphate buffered saline (PBS) and fixed in 4% Formaldehyde (Sigma F8775) for 20 min at room temperature (RT). For Baz-PS180 phosphostainings (*Figure 3*, *Figure 3—figure supplement 1*), tissues dissected in PBS and fixed in 2% Trichloroacetic acid for 3 min at RT. Regardless of the fixation procedure, tissues were then permeabilised in for 1 hr in PBS-Triton 0.1% (PBT) at RT, and incubated overnight at 4 °C in a solution of primary antibody diluted in PBT. They were rinsed twice and washed for 1 hr in PBT, incubated 1 hr at RT in a solution of DAPI 1/1000 and fluorophore-coupled secondary antibodies (Thermo Fisher) diluted in PBT, rinsed twice and washed in PBT, rinsed twice in PBS, rinsed in 50% glycerol (Sigma 49781) and finally mounted between glass and coverslip in Vectashield (Vector Laboratories H-1000). Primary antibodies used were as follows: rabbit anti-PKC ζ (Santa Cruz, sc-17781, 1:500); Sheep-anti-Mira (1:1000); Rabbit anti Mira; Sheep anti-Numb (1:500); Mouse anti-Cyclin A (DSHB A12 1/500); Mouse anti-Su(H) (Santa Cruz sc-398453 AF647, 1/1000); Mouse anti-Cut (DSHB 2B10 1/200), Sheep anti-Baz-pS180 (this study, 1/200).

## Live imaging of larval brains

Every reference to Schneider's medium corresponds to glucose-supplemented (1 g l⁻¹) Schneider's medium (SLS-04–351Q). Tissue mounting and live imaging were performed as described (*Januschke and Loyer, 2020*). Entire brains were dissected from L3 larvae in Schneider's medium and isolated from the surrounding imaginal discs. Particular care was taken to avoid pulling on brains at any time during the dissection and damaged brains were discarded. Isolated brains were transferred to a drop of Fibrinogen dissolved in Schneider's medium (10 mg/ml) on a 23 mm Glass bottom dish (WPI), which was then clotted by addition of Thrombin (100 U/ml, Sigma T7513). Clots were then covered in Schneider's medium, taking into account the existing volumes of Schneider +Fibrinogen and Thrombin forming the clots for a final volume of 200 µl. For experiments involving 1-NA-PP1 addition, control and analog-sensitive tissues were mounted on different clots in the same dish and 1-NA-PP1 was added during imaging by adding 200 µl of Schneider with double the desired final concentration of 1-NA-PP1 (*e.g.* 200 µl of 20 µM 1-NAPP-1 in Schneider medium were added to the coverslip for a final concentration of 10 µM). Where applicable, neuroblasts were arrested in metaphase by exposure to 100 µM Colcemid (Sigma 234109).

## Live imaging of embryonic neuroblasts

Embryos were collected 3–4 hr after egg laying, manually dechorionated by rolling them on double-sided tape and transferred to a well containing Schneider's medium. The posterior tip of the eggshell was nicked with a sharp forceps and the embryos were squeezed out by pressing from the anterior to the posterior of the eggshell. The mounting in Fibrin clots, live imaging and exposure to 1-NA-PP1 of de-shelled embryos then proceeded as described above for larval brains, except for the imaging medium, this time supplemented with glucose (1 g l−1), Insulin (75 µg/ml, Lonza, BE02-033E20), 10% Fetal Calf Serum and 2.5% fly extract (*Drosophila* Genomics Resource Center, 1645670).

## Live imaging of pupal nota

Pupae were removed from their pupal case and were stuck on a 23 mm Glass bottom dish (WPI) with heptane glue (made by incubating 20 4 cm pieces of Scotch 'Magic' tape [3 M] in 10 ml of heptane overnight). A wet tissue was placed inside the dish to maintain moisture throughout imaging.

## Cloning and transgenesis

The analog sensitive allele *cdk1*[as2] was generated by genome editing using the guide RNA GATC TTTGAATTCCTATCGA TGG and a template introducing the F80A (TTT to GCC) mutation and a

silent mutation at L83 (CTA to TTG) to prevent re-editing by the guide RNA. The UASz-driven GFP-tagged Baz transgenes correspond to the Baz-PA isoform, in which 6xHiseGFP was introduced at the same position (between K40 and P41) and with the same linker peptides as the functional Baz::GFP protein trap (*Buszczak et al., 2007*) and Baz::mScarlet-I knock-in (*Houssin et al., 2021*) using Gibson assembly cloning into UASz 1.0 (DGRC 1431). To generate the non-phosphorylatable and phospho-mimetic constructs in the relevant Serines in Baz, site-directed mutagenesis was used to introduced alanine (GCG) or aspartic acid (GAG), respectively. All vectors were verified by sequencing (vector sequences available upon request). Transgenic lines were generated by site-directed transgenesis (https://www.flyfacility.gen.cam.ac.uk/Services/Microinjectionservice) into attP2.

## Image processing and analysis

All images were processed and filtered using ImageJ. For all images, we applied a 3D gaussian blur with X, Y and Z sigmas of 0.8 pixels. For all images where applicable, we applied a Gamma filter with a value of 0.5 on the DAPI or Histone::RFP channels. For the timing of NEB, we considered that NEB has started as soon as the Baz signal is no longer excluded from the nucleus. Cytoplasmic signal intensity was measured inside polygonal shapes excluding the cortex, the nucleus or chromosomes. The background fluorescence was measured outside of brains and subtracted from any measured signal. Measurement of cortical signal intensity was performed as described before using our custom rotating linescans macro (*Januschke and Loyer, 2020*). Briefly: (**1**) in ImageJ, the user traces a line selection roughly perpendicular to the cortex they want to measure. (**2**) The user increases the line width as much as possible while keeping the intersection between the line selection and the cortex a straight line. (**3**) The user starts the rotating linescans macro, which perform linescans (i.e. measuring signal at each pixel along the length of line while averaging pixels values along the width of the line) along the wide line selection traced by the user. For each of these linescans, the line selection is rotated with a 1° step from –25° to +25° relative to the original orientation of the line selection. (**4**) The maximal intensity measured across all different orientations corresponds to the orientation where the line is perfectly perpendicular to the cortex. For this orientation, the macro measures the pixel values at the intersection between the line selection/cortex and 1 pixel around this intersection. (**5**) The macro performs steps 3 and 4 one optical slice above and one optical slice below the optical slice initially selected by the user. (**6**) The macro returns a final intensity value corresponding to the average of the three values measured as described in step 4 on each optical slice.

## Quantitative data representation and statistical analysis

For every boxplot: cross: maximal and/or minimal outliers (beyond 1.5×interquartile range); grey circle: average; red dots: individual measurements; centre line, median; box limits, upper and lower quartiles; whiskers, 1.5×interquartile range. p Values were calculated using a non-parametric two-tailed Mann–Whitney U test in all cases, except for *Figure 5G* and *Figure 6—figure supplement 1H′*, which correspond to a two-tailed Z score calculation for two population proportions.

## Purification of Baz and PARD3 N-terminal fragments

Fragments were expressed and purified by standard approaches. Briefly, 2–4 L of BL21(DE3) cells were harnessed to express the relevant fragments at 18 C overnight following IPTG induction at OD600=0.8. Following harvest, cells expressing GST-Par3/Baz truncations were thawed on ice and homogenised in Standard Buffer (20 mM HEPES, 250 mM NaCl, 0.5 mM TCEP, pH 7.4). 1 x Protease inhibitor cocktail and 5–10 µL Benzonase were added. Samples were lysed by sonication and centrifuged at 60,000 x $g$ at 4 °C for 1 hr. Two mL Glutathione Resin (MRC Reagents and Services) was preequilibrated in Standard Buffer. The supernatant was filtered through a 0.22 µm filter using a syringe and mixed with the pre-equilibrated resin, and incubated on a rotary spinner at 4 °C for 1 hr. Samples were centrifuged at 500 x $g$ at 4 °C for 5 min and the flow-through was collected. Forty mL of Buffer 1 was added to the resin, and it was centrifuged at 500 x $g$, 4 °C for 10 min and the excess buffer removed. This wash step was repeated another 2 times. Resin with sample was incubated in 10 mL Standard Buffer supplemented with 10 mM Glutathione to elute the protein. The protein in the supernatant was collected after centrifuging at 500 x $g$ at 4 °C for 10 min. Subsequently, the protein elution from the affinity chromatography was injected onto a HiLoad 16/60 Superdex 200 prep grade sizing column (Fisher, #15182085) pre-equilibrated in Standard Buffer and peak fractions

were collected. Samples were analysed by SDS-PAGE and Coomassie gel stain. Protein fractions were identified, aliquoted snap frozen using LQN2 and stored at –80 °C.

## Immunoblotting

Samples were stored at –20 °C until SDS-PAGE electrophoresis and immunoblotting. NuPAGE 4–12% SDS-PAGE gels were transferred onto nitrocellulose membrane and incubated with primary antibody diluted in 1 X TBS-T 5% BSA overnight in 4 °C. Phosphospecific anti-baz pS180 antibody was used at 0.2 µg/mL with 10 µg/mL non-phosphopeptide (S180) to decrease the non-specific background. After incubation with primary antibodies, membranes were washed 3 times with TBS-T buffer (20 mM Tris-HCl (pH 7.5), 150 mM NaCl supplemented with 0.2% (v/v) Tween-20 (Sigma Aldrich)) and incubated for 1 hr in secondary antibodies (1:10000 dilution in 1 X TBS-T 5% BSA) at room temperature, membranes were washed three times in TBS-T and subjected to Infrared detection using LI-COR Odyssey Clx system.

## In vitro kinase phosphorylation assay

A total of 2.5 µg of recombinant GST-tagged *Drosophila* Bazooka (2-320) or GST-tagged human PAR3 (2-281) was incubated in the presence or absence of 200 ng human GST-CDK1/GST-Cyclin B1 in a kinase reaction for 60 min at 30 °C. Samples were subjected to S-TRAP micro high recovery protocol with one short trypsin digest.

## p-MS MASCOT

Phosphoproteomic analysis by liquid chromatography mass spectrometry (LC-MS/MS). LC separations were performed with a Thermo Dionex Ultimate 3000 RSLC Nano liquid chromatography instrument using 0.1% formic acid as buffer A and 80% acetonitrile with 0.08% formic acid as buffer B. The peptide samples were loaded on C18 trap columns with 3% acetonitrile / 0.1% trifluoracetic acid at a flow rate of 5 µL/min. Peptide separations were performed over EASY-Spray column (C18, 2 µM, 75 µm x 50 cm) with an integrated nano electrospray emitter at a flow rate of 300 nL/min. Peptides were separated with a 60 min segmented gradient starting from 3% to 7% buffer B over 5 min, 7%~25% buffer B over 43 mins, 25%~35% buffer B over 5 min, then raised to 95% buffer B over 3 min and held for 2 min. Eluted peptides were analysed on an Orbitrap Exploris 480 (ThermoFisher Scientific, San Jose, CA) mass spectrometer. Spray voltage was set to 2 kV, RF lens level was set at 40%, and ion transfer tube temperature was set to 275 °C. The mass spectrometer was operated in data-dependent mode (ID low load) with 2 seconds per cycle. The full scan was performed in the range of 350—1200 m/z at nominal resolution of 60,000 at 200 m/z and AGC set to 300% with a custom maximal injection time of 28ms, followed by selection of the most intense ions above an intensity threshold of 10000 for higher-energy collision dissociation (HCD) fragmentation. HCD normalised collision energy was set to 30%. Data-dependent MS2 scans were acquired for charge states 2–6 using an isolation width of 1.2 m/z and a 30 s dynamic exclusion duration. All MS2 scans were recorded with centroid mode using an AGC target set to standard and a maximal fill time of 100ms.

The.RAW files obtained from the mass spectrometer were processed by Proteome Discoverer v2.4 (ThermoFisher Scientific) using Mascot v 2.6.2 (Matrix Science) as the search engine. A precursor mass tolerance of 10ppm and fragment tolerance of 0.06 was used. An in-house database (MRC_Database_1) was used with trypsin set as the protease which was allowed a maximum of two missed cleavage sites. Oxidation and dioxidation of methionine, phosphorylation of serine, threonine and tyrosine were set as variable modifications, and carbamidomethylation of cysteine was set as a fixed modification. Phospho-site assignment probability was estimated via Mascot and PhosphoRS3.1 (Proteome Discoverer v.1.4-SP1) or ptmRS (Proteome Discoverer v.2.0). ptmRS was used as a scoring system for the phospho site identification, with a mass tolerance of 0.5 and neutral loss peaks were considered. Phosphorylation site localisation was considered correct only if peptides had a Mascot delta score and ptmRS probability score above 85%.

## Acknowledgements

We thank C Gonzalez and M Krahn for the gift of flies. We thank the Dundee imaging facility for support. Stocks obtained from the Bloomington *Drosophila* Stock Center (NIH P40OD018537) were used in this study. We thank R Le Borgne for discussion and C Roubinet for critical reading of the

manuscript. Work in JJ's was supported by a fellowship from Wellcome (100031/Z/12/A) and grants from the BBSRC (BB/V001353/1 and BB/T017546/1). GMF and EH were supported by a Wellcome Trust/Royal Society Sir Henry Dale Fellowship (211209/Z/18/Z). DHM was supported by a Wellcome Trust/Royal Society Sir Henry Dale Fellowship (211193/Z/18/Z) and a Royal Society (RGS\R2\180284) grant.

## Additional information

### Funding

| Funder | Grant reference number | Author |
| --- | --- | --- |
| Wellcome Trust | 10.35802/100031 | Jens Januschke |
| Wellcome Trust | 10.35802/211193 | David H Murray |
| Wellcome Trust | 10.35802/211209 | Greg M Findlay |
| Biotechnology and Biological Sciences Research Council | BB/V001353/1 | Jens Januschke |
| Royal Society | RGS/R2/180284 | David H Murray |
| Biotechnology and Biological Sciences Research Council | BB/T017546/1 | Jens Januschke |

The funders had no role in study design, data collection and interpretation, or the decision to submit the work for publication. For the purpose of Open Access, the authors have applied a CC BY public copyright license to any Author Accepted Manuscript version arising from this submission.

### Author contributions

Nicolas Loyer, Conceptualization, Data curation, Formal analysis, Investigation, Methodology, Writing - original draft, Writing - review and editing; Elizabeth KJ Hogg, Hayley G Shaw, Anna Pasztor, Data curation, Investigation, Methodology; David H Murray, Conceptualization, Supervision, Methodology; Greg M Findlay, Supervision, Investigation, Methodology; Jens Januschke, Conceptualization, Data curation, Formal analysis, Supervision, Funding acquisition, Methodology, Writing - original draft, Project administration, Writing - review and editing

### Author ORCIDs

Nicolas Loyer ⬚ http://orcid.org/0000-0001-5010-2564
Elizabeth KJ Hogg ⬚ http://orcid.org/0000-0002-2509-3202
Anna Pasztor ⬚ http://orcid.org/0009-0001-4659-9691
David H Murray ⬚ http://orcid.org/0000-0003-2582-8552
Greg M Findlay ⬚ http://orcid.org/0000-0002-7222-4965
Jens Januschke ⬚ http://orcid.org/0000-0001-8985-2717

### Decision letter and Author response

Decision letter https://doi.org/10.7554/eLife.97902.sa1
Author response https://doi.org/10.7554/eLife.97902.sa2

## Additional files

### Supplementary files

• Supplementary file 1. Phosphopeptides identified from phospho-proteomics of *Drosophila* Baz and human PARD3 N-terminal fragments in the presence or absence of CDK1/CyclinB1. Bold text: very good position assignment. Red text: phosphorylation also observed in the absence of CDK1/CyclinB1.

• Supplementary file 2. Genotypes of *Drosophila* lines used in this study.

• Supplementary file 3. Origin of *Drosophila* stocks used in this study.
• MDAR checklist

## Data availability

The raw western blot data shown in *Figure 7* are available in *Figure 7—source data 1*.The mass spectrometry proteomics data presented in *Figure 7* have been deposited to the ProteomeXchange Consortium via the PRIDE partner repository with the dataset identifier PXD051560.

The following dataset was generated:

| Author(s) | Year | Dataset title | Dataset URL | Database and Identifier |
|---|---|---|---|---|
| Hogg EKG, Findlay G | 2024 | A CDK1 phosphorylation site on *Drosophila* PAR-3 regulates neuroblast polarisation and sensory organ formation | https://www.ebi.ac.uk/pride/archive/projects/PXD051560 | PRIDE, PXD051560 |

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
