## [Editor Report]

The coupling of polarity to the cell cycle is critical to ensuring polarization, spindle position and fate asymmetries are properly linked to cell cycle progression. Using a combination of an analog sensitive Cdk1 allele and phoshomimetic/non-phosphorylatable mutants, this important work convincingly shows the impact of Cdk1 on polarity domain coalescence, Baz/Par3 localisation, and fate specification that are of broad interest to the field.

---

## [Decision Letter]

[Editors' note: this paper was reviewed by Review Commons.]

---

## [Author Response]

*General Statements [optional]*

Title: we have changed the title from:

“Asymmetric cell division-specific phosphorylation of PAR-3 regulates neuroblasts polarisation and sensory organ formation in *Drosophila*”

To:

“A CDK1 phosphorylation site on *Drosophila* PAR-3 regulates neuroblast polarisation and sensory organ formation”

Abstract:

We have added the following findings to the abstract:

CDK1 phosphorylates Baz-S180 and human PARD3-S187 in vitroBaz-S180 phosphorylation affects its localisation during the asymmetric cell division of sensory organ precursorsWe toned down our claim that Baz-S180 phosphorylation is specific to asymmetric cell division

Results (see below for the corresponding figures)

We now describe the Numb phenotype in the Baz phosphomutants experiments in Baz phosphomutants (Figure 4G-H’, Reviewer 2 request)We now show that Baz-pS180 is phosphorylated in the early pupal notum during lateral inhibition (Figure 5H, request from both reviewers)We now show that Baz-pS180 phosphorylation controls its localisation during sensory organ precursors divisions (Figure 6, Reviewer 1 request).We describe mosaic expression of UASz-driven Baz within the SOP lineage, how we used it to analyse the contribution of SOP lineage cells to various pools of cortical Baz and how it revealed a new cell non autonomous mechanism controlling the pIIb cell division orientation (Figure S6).We replaced the last part of our initial submission (“Human PARD3 is a substrate of CDK1/CyclinB1 in vitro”, associated to Figure 6) with a new part (“*Drosophila* Baz-S180 and human PARD3-S187 are phosphorylated by CDK1/CyclinB1 in vitro”, associated to Figure 7 and a new Supplementary Table 1), in which we purify Baz and PARD3 N-terminal fragments and show that they are phosphorylated in vitro by CDK1 on S180 and S187, respectively.

Discussion

We now discuss the above findings (except for those related to Figure S6, which are side observations that should be of interest to SOP specialists).We attempted to simplify the part discussing the Notch/Baz phenotypes following Reviewer 2’s comment.

Material and Methods

We describe the methods related to the above findings.We now detail our method for cortical signal measurement with a custom macro (Reviewer 2 request).We now describe boxplots in the Material and Methods and in the legend in the first instance they are used (Figure 2A’, Reviewer 2 request).

Changes in figures

**Author response table 1. sa2table1:** 

Figure 1	No change
Figure 2	• Panel A now also shows a control neuroblast (Reviewer 2 request)• Panels after G have been rearranged to put the quantification of the Mira signal (now 2J) next to the Numb signal (now 2K, Reviewer 2 request)
Figure S2	A new panel (B) shows two successive divisions of a cdk1AS2 neuroblast in the absence of 1-NAPP1 (Reviewer 1 request)
Figure 3	No change
Figure S3	No change
Figure 4	New panels (G-H’) now describe the Numb phenotype in the Baz phosphomutants experiments (Reviewer 2 request)
Figure 5	The caption in Figure 5A now indicates that RFP RNAi depletes Baz::mScarletA new panel (H) now shows that Baz-pS180 is phosphorylated in the early pupal notum during lateral inhibition (Request from both reviewers)
Figure 6	The previous Figure 6 has been replaced with Figure 7 (see below). This entirely new Figure 6 shows that Baz-S180 phosphorylation affects Baz localisation during the pIIa cell asymmetric cell division in the SOP lineage (Reviewer 1 request)
Figure S6	This entirely new figure describes mosaic expression of UASz-driven Baz within the SOP lineage and how we took advantage of it to analyse the contribution of SOP lineage cells to various pools of cortical Baz. It also describes a new cell non autonomous mechanism controlling the pIIb cell division orientation.
Figure 7	We have replaced the previous Figure 6 (testing full length human PARD3 phosphorylation by CDK1 in vitro) with this entirely new Figure 7. We synthesized *Drosophila* Baz and human PARD3 N-terminal fragments and showed that CDK1 phosphorylates Baz-S180 and the “equivalent” PARD3-S187 in vitro.

Reviewer #1 (Evidence, reproducibility and clarity (Required)):The coupling between cell polarity and cell cycle progression is an important aspect of symmetric and asymmetric cell division. Although there are several examples of cell cycle kinases phosphorylating polarity proteins, it has been difficult to assess the importance of these on cell division due to the strong and pleiotropic effects of manipulating these kinases. Here, the authors generate an analogue-sensitive allele of cdk1 in flies to tackle this question in neuroblasts (NBs) and sensory organ precursors (SOPs), two well characterised examples of asymmetric cell divisions. They show that partial Cdk1 inhibition (which still allows cell cycle progression) does not block Bazooka (PARD3 in mammals) polarization in NBs, but prevents coalescence of the Baz crescent, which has previously been shown to be an actomyosin-based process. They further identify a Cdk1 consensus site on Baz (S180) for which they generate a phospho-specific antibody, allowing them to show that this site is specifically phosphorylated in dividing NBs and SOPs. Although mutations at this site do not recapitulate the effect of Cdk1 on Baz coalescence, they do delay Miranda polarization in NBs and affect lateral inhibition and asymmetric cell division of SOPs. Finally, the authors show that human PARD3 can also be phosphorylated by Cyclin B/Cdk1 in vitro.Major comments:1. Figure 2A: it would be good to show that polarization of Baz::GFP in consecutive divisions is maintained in cdk1as2 animals in the absence of 1-NA-PP1.

We now show in Figure 1 —figure supplement 1B a panel with two consecutive divisions of a *cdk1^as2^* neuroblast in the absence of 1-NAP-PP1, followed by a third division in the presence of 1-NAP-PP1. The neuroblast shows high levels of Baz polarization in the two first divisions.

2. The interpretation of the observed SOP phenotypes is complicated by the uneven expression of the pnr-GAL4 driver and the fact that it is expressed in epithelial cells rather than just SOPs. The authors could express their control and mutant Baz constructs under the control of neurP72-GAL4. It is not likely they would be able to deplete endogenous Baz as they have done in NBs, as neurP72-GAL4 is expressed too late to deplete most proteins before SOP division, but they could at least look at localization of the mutants and any possible gain-of-function phenotypes.

Following this suggestion, we have recombined Neur-*GAL4* with UAS-*δ RNAi* to attempt to deplete both endogenous Baz::mScarlet and Δ while expressing our Baz::GFP constructs specifically in SOPs. Baz::mScarlet depletion was surprisingly efficient considering, as the reviewer points out, the late timing of Neur-GAL4 expression. However, the adult flies did not present any sensory organs transformations, perhaps because Δ might not be as efficiently depleted. We can at least rule out dominant-negative effects.

We thank the reviewer for his constructive feedback and as suggested, we now extensively analysed the localisation of the Baz-S180 mutants in SOPs and found significant defects. We describe these observations in a new Figure 6. Briefly, we observed that the Baz phosphomutants have localisation defects during the pIIa cell division but not the π cell division. We also observed a very surprising mosaicism of expression of our UASz-driven constructs within the SOP lineage that allowed us to make a few interesting observations which should be of interest to SOP specialists. Briefly, mosaic expression of Baz::GFP within the SOP lineage allowed to analyse the relative contributions of pIIa and pIIb/pIIIb to different Baz cortical pools and revealed an unexpected cell non-autonomous mechanism controlling pIIb division orientation. We describe these findings in a new associated supplemental figure.

3. The authors speculate that Baz phosphorylation during lateral inhibition may be the reason for the observed excess specification of SOPs in the S180 mutants. This could easily be tested by looking at their antibody staining at earlier stages in the notum.

Following this suggestion (also coming from Reviewer #2), we have stained nota between around 8h APF. We observed that patches of cells of the early notum display a strong Baz-pS180 phospho-signal. These patches partially overlap with the Δ-positive stripes in which lateral inhibition occurs (as described for example in Corson et al., 2017), consistent with the possibility that Baz-S180 phosphorylation does somehow regulate lateral inhibition.

These new experiments clearly show that Baz can be phosphorylated on S180 in cells that do not divide asymmetrically. This led us to change the title.

Reviewer #1 (Significance (Required)):This work advances our knowledge of the coupling between the cell cycle and cell polarity during cell division, and shows that Baz/PARD3 receives inputs from Cdk1 that is specific to asymmetrically dividing cells. The reagents generated here (cdk1as2 and phospho-specific antibody) will also be of interest to the field. The data are convincing and well documented. This work should be of broad interest to the stem cell and developmental biology fields. Above are a few suggestions to improve the manuscript.Reviewer #2 (Evidence, reproducibility and clarity (Required)):Cell polarization in dividing cells, including stem cells, is typically coupled such that polarity can inform the architecture, orientation, and/or asymmetry during cell division. In *Drosophila* neural stem cells (neuroblasts/SOP), Par polarity is coupled to the cell cycle, but the nature of this coupling remains unclear. In this work, Loyer and colleagues report on impacts of CDK1 inhibition on Bazooka/Par3 localization and basal fate determinant localization. They provide evidence for a novel phosphorylation site that appears unique to asymmetrically dividing cells and may be involved in regulation of asymmetric division. Finally, they show that CDK1 can, at least in principle, phosphorylate human Par3 in vitro.Overall, the major claims of the abstract appear supported by the experimental work; however, we think the title overstates the overall conclusions that can be drawn from the work.Major comments:(1) The major claim of the paper is the role of specific phosphorylation of S180 in asymmetrically dividing cells in polarization and sensory organ formation, which relies heavily on interpretation of S180A/D phosphomutants. The experiments are carefully performed and quantified, and are consistent with the conclusions drawn. However, we wondered if it possible that the phenotypes are not linked to phosphorylation (the authors acknowledge this in the Discussion)? In other words could the A/D mutants simply be weak Baz mutants? This could this potentially explain the extra-SOP phenotype if Baz function is generally altered, especially given that it is difficult to rationalise a role for SOP-specific phosphorylation in the processes that specify SOP cells in the precursor epithelial cells. The authors speculate that these early precursors may exhibit also phosphorylation, but this isn't examined. Chasing this down seems key to support the core titular claim of the paper.

Following this suggestion (also coming from Reviewer #1), we have stained nota around 8h APF. We observed that patches of cells of the early notum display a strong Baz-pS180 phospho-signal. These patches partially overlap with the Δ-positive stripes in which lateral inhibition occurs (as described for example in Corson et al., 2017). This result is presented in Figure 5H. As would be the case for any phosphomutant, this does not strictly rule out that the S180A and S180D could simply be weak Baz mutants, but it strongly supports the possibility that the lateral inhibition defects observed in these mutants result from defective Baz-S180 phosphorylation.

(2) Implicit in the core message of the paper is the elucidation of CDK1 regulation of polarity and specifically Baz. However, the connection between CDK1 and S180 (and Baz regulation overall) is relatively tenuous in this work. First, the S180A mutant does not phenocopy CDK1 inhibition with respect to basal determinant phenotypes, though obviously CDK1 may be more pleiotropic. Second, whether the CDK1 inhibition phenotype is linked to any effect on Baz/PAR behaviour is not really explored. Third, they do not test whether S180 phosphorylation is CDK1-dependent.

We fully agree with these comments. We cannot think of any way of addressing the first two points, which would require fully inhibiting CDK1 and somehow maintaining neuroblasts in mitosis to examine how it impacts Baz localisation. We tried to arrest neuroblasts in mitosis and block the proteasome as this at least in HeLa cells led to persistence of mitosis when CDK1 was inhibited (Skoufias et al., 2007). However, neuroblasts arrested in mitosis by proteasome inhibition slipped out of mitosis.

However, concerning the third point, we now provide evidence showing that, at least in vitro, *Drosophila* BazS180 is phosphorylated by CDK1 (see below).

(3) The method for quantifying domain signal only references prior work and should be described in this work. From our search of the cited reference, it appears to be peak signal intensity at a user specified point on the cortex. While this does not undermine the core findings as presented, it may not capture additional features that may be informative (domain size, fluorescence distribution, total signal etc.). For example domain coalescence would imply smaller, brighter domains, but similar total protein amounts, which appears to be the case from images, but isn't quantified per se.

We now describe our method for quantifying average signal intensity in the middle of the Baz crescents. We agree that quantifying additional features to check whether they are affected by partial CDK1 inhibition would be interesting. However, doing so requires determining exactly where Baz crescents start and end. As Baz crescent edges in neuroblasts often end in a gradient rather than a sharp edge (Hannaford et al., 2018), we are not sure to be able to confidently do so in every case with the image quality of our dataset: we prioritised limiting photobleaching to accurately quantify the levels of endogenously expressed Baz rather than obtaining very sharp and high contrast images. This is further complicated by the fact that, depending on the depth of neuroblasts within the tissue and the orientation of their division relative to the imaging plane, the signal intensity of Baz crescents is quite variable, preventing a simple thresholding approach to arbitrarily determine the limits of crescents based on signal intensity. In short, accurately determining the size of crescents is very challenging.

(4) The phosphospecific antibody signal is relatively weak, leading to relatively low signal to noise, which could compromise the ability to detect phospho-S180 in non-asymmetrically dividing cells or generally in cells in which Baz is not polarised and thus signal would be diffused around the cell rather than concentrated. Similar caveats could also apply to the lack of signal in interphase cells, where Baz may be less enriched at the cortex and not polarized. We are inclined to believe the authors conclusions, particularly given their examination of multiple cell types and tissues. However, it is a potential caveat as it may be most visible in polarised cells where it is asymmetrically enriched.

We thank the reviewer for pointing this out. Given the fact that Baz levels at the neuroepithelial cells adherens junctions are similar, we are confident that Baz-S180 is phosphorylated in dividing neuroblasts but not in non-mitotic epithelial cells, which is at least consistent with our new finding that CDK1 phosphorylates Baz-S180 in vitro. However, we agree that we cannot strictly rule out that Baz-S180 is phosphorylated but below a detection threshold in mitotic neuroepithelial cells as cortical Baz levels decrease in these cells.

We have also gathered new data showing that, in the early notum, Baz-S180 is detected in epithelial cells that are not dividing asymmetrically, definitely ruling out the notion that Baz-S180 is strictly ACD-specific. We have changed the title of the paper accordingly, toned down the mention of apparently ACD-specific Baz-S180 phosphorylation in the abstract and now describe and discuss the fact that the apparent ACD-specificity of Baz-S180 phosphorylation is context-specific.

(5) Examination of in vitro phosphorylation of human Par3D (Figure 6) seems out of place and does not add much. It is human, not Bazooka. They reveal 30 sites, 18 of which in both replicates, but most are not obvious CDK sites and the S180 equivalent site is missing. None of these sites is validated in vivo, at least in this work.

We fully agree with these comments. We initially attempted to purify both full length Baz and human PARD3 but only managed to purify small amounts of PARD3, which is why our analysis was limited to human PARD3. To circumvent these difficulties, we instead purified a smaller N-terminal fragment of Baz and PARD3, which was successful for both proteins and gave us much higher quantities of sample for analysis. Using two different approaches (Western blot with our phospho-specific antibody on Baz and targeted mass spectrometry on Baz and PARD3), we now show in a new Figure 7 that CDK1 phosphorylates Baz-S180 and PARD3-S187 in vitro.

Minor comments:– Figure 1: Uses metaphase arrested cells, presumably colcemid, but colcemid is only noted in Figure 2.

We now mention Colcemid in the legend of Figure 1.

– Figure 2A: Scale bar is truncated.

We have corrected this.

– Figure 2A: Example images of control neuroblasts could be useful to readers.

We now show control neuroblasts in Figure 2A.

– Figure 2G' vs H': Because G' has two panels and H' has only one, we often confused the PKC and Mira box plots when comparing to Numb. Perhaps Mira could be in a separate sub panel or be more closely juxtaposed with Numb?

The quantification of the Mira signal is now right next to Numb.

– Whereas both Numb/Mira were examined in CDK1(as), only Mira is reported for the S180A/D experiments. Is there a Numb phenotype as well?

We actually co-stained Numb and Miranda in the dataset that we analysed in the S180A/D experiments shown in Figure 4E, F. We did not analyse Numb localisation in the first version we submitted because of a penetration issue of the Numb antibody: the Numb signal fades extremely fast as we image deeper in the tissue, causing large difference of signal intensity even within a single cell. This prevents us from performing any meaningful quantitative measurement of the Numb signal like the one we did in Figure 2H, K, for which we did not encounter this issue. All our further immunostaining experiments with this antibody have had the same problem since then, even after using Triton concentrations up to 4% for permeabilization.

Nonetheless, following the reviewer’s question, we have at least performed a simple qualitative analysis of Numb localisation in this experiment. We observed that Numb localised to the basal pole in most cases in controls and Baz phosphomutants, but localised uniformly at the cortex in half the cases where Miranda showed very low levels of polarisation in metaphase in Baz^S180D^ mutants. This Numb localisation defect suggests a loss of function of the PAR complex whereas, intriguingly, the Miranda localisation defect suggests a gain of function of the PAR complex. These new observations are described in Figure 4G-H’.

– The discussion of the notch / Baz phenotypes (Figure 5) is rather complicated and a bit difficult to follow.

We agree with this, we have rewritten this part. This is further simplified by our new observation that Baz-S180 is phosphorylated in the early notum during lateral inhibition.

– Figure 5A: captions should indicate that RFP RNAi is depleting Baz.

We have modified the figure accordingly.

– Box plots are used, but not described. i.e. outliers seem to be marked, but criteria unclear. Mean vs median, etc.

We now describe boxplots in the legend in the first instance they are used (**Figure 2A’**), and in the material and methods

– Some grammatical mistakes:– Title: neuroblast (no 's'),– Page 1: Cell fate difference(s?) in the resulting daughter cells– Page 4: (As) CDK1 inhibition with 10 μM 1-NA-PP1 prevents neuroblasts from cycling and causes metaphase- arrested neuroblasts to slip out of mitosis. (Reword)– Page 6: increased levels of basal fate(no 's') determinants

We have corrected these mistakes.

Reviewer #2 (Significance (Required)):The links between cell cycle and cell polarity are clearly important and remain poorly understood. Hence, the work addresses key conceptual/mechanistic questions relevant to our fundamental understanding of stem cell biology and regulation of polarity and asymmetric cell division. In our opinion, there are clearly some interesting observations in the manuscript, the experiments are performed carefully, and the data are generally well described. That said, overall, the work seems somewhat premature.(1) The direct impact of CDK1 on Baz behaviour remains somewhat unclear. The authors do a good job of limiting the concentration of inhibitor to decouple effects of cell cycle progression from CDK1 levels per se, but this does potentially impact the strength of the phenotypes they can detect and hence the observed phenotypes are relatively minor. Note that driving cells out of mitosis with stronger CDK1 inhibition clearly impacts Baz localization, so the 'real' effect of CDK1 inhibition on Baz could be stronger than reported here. It is also unclear whether the phenotypes observed are directly linked to CDK1 regulation of PAR polarity or an indirect effect of cell cycle control of other processes. The authors' suggestion that it could be related to defects in cortical actin organization, which is known to be cell cycle controlled, seems most likely, but neither this or other models are explored further.

We agree but are not aware of any experiment that would allow testing full inhibition of CDK1 on membrane-bound Baz in mitotic neuroblasts. As mentioned above in our response to reviewer #1 we tried to arrest neuroblasts in mitosis and block the proteasome as this at least in HeLa cells led to persistence of mitosis when CDK1 was inhibited (Skoufias et al., 2007). However, neuroblasts arrested in mitosis by proteasome or Colcemid or both slipped out of mitosis upon inhibition of CDK1.

We agree it would be interesting to study how CDK1 affects the actomyosin network in neuroblasts but feel that this is somewhat beyond the scope of the manuscript.

(2) Using phosphospecific antibodies, they report on a novel putative CDK1 phosphorylation site, but aside from looking like a consensus CDK1 site, whether this site is CDK1 dependent is not examined. Notably, the corresponding phosphomutants have modest effects and don't obviously account for the CDK1 inhibition phenotype, leaving it somewhat unclear whether it is under cell cycle regulation.

We now provide a new Figure 7 to address this point. As mentioned already above, using two different approaches (Western blot with our phospho-specific antibody on Baz and targeted mass spectrometry on Baz and PARD3 using), we now show in a new Figure 7 that CDK1 phosphorylates Baz-S180 and PARD3-S187 in vitro. Again, we cannot identify any experiment that would allow us testing whether S180 Baz is a direct target of CDK1 in vivo. The fact that we now report significant defects on Baz localisation in pIIa divisions, strongly suggests functional relevance and CDK1 seems a plausible kinase based on the new in vitro results.

(3) The observation that S180 phosphorylation appears unique to asymmetrically dividing cells is very curious, but this observation is not followed up extensively. Again phenotypes of phosphomutants are quite modest, and while one can propose models to rationalise the phenotypes observed, these models are not fully explored.

As mentioned above, we now show that Baz-S180 phoshorylation is not strictly ACD-specific and changed the title accordingly. We also have new data showing that the S180 phosphomutants of Baz have localisation defects in mitotic pIIa divisions (new Figure 6). Therefore, this phosphorylation event on Baz can be linked to Baz’s cortical localisation and interestingly shows context dependency.

(4) The findings that human Par3D can be phosphorylated by CDK1 in vitro do not add much particularly as they obtain a very large number of putative sites raising questions of specificity, the sites are not validated, and an S180 equivalent site was not identified.

We agree that this has been a weakness which we feel we have addressed. We paste here the answer already provided above when replying to reviewer #1.

We initially attempted to purify both full length Baz and human PARD3 but only managed to purify small amounts of PARD3, which is why our phospho-proteomics analysis was limited to human PARD3. To circumvent these difficulties, we instead purified a smaller N-terminal fragment of Baz and PARD3, which was successful for both proteins and gave us much higher quantities of sample for analysis. Using two different approaches (Western blot with our phospho-specific antibody on Baz and phosphor proteomics on Baz and PARD3 using mass spectrometry), we now show in a new Figure 7 that CDK1 phosphorylates Baz-S180 and PARD3-S187 in vitro.

In summary, the individual findings of this work are interesting and generally solid. Each could be followed up to provide mechanistic insight into cell cycle- or cell type-dependent regulation of Par polarity. However, in their current state, the results seem more like a loosely connected set of observations.Expertise: Cell polarity and asymmetric cell division

References

CORSON, F., COUTURIER, L., ROUAULT, H., MAZOUNI, K. & SCHWEISGUTH, F. 2017. Self-organized Notch dynamics generate stereotyped sensory organ patterns in *Drosophila*. *Science,* 356.

HANNAFORD, M. R., RAMAT, A., LOYER, N. & JANUSCHKE, J. 2018. aPKC-mediated displacement and actomyosin-mediated retention polarize Miranda in *Drosophila* neuroblasts. *eLife,* 7**,** 166.

SKOUFIAS, D. A., INDORATO, R. L., LACROIX, F., PANOPOULOS, A. & MARGOLIS, R. L. 2007. Mitosis persists in the absence of Cdk1 activity when proteolysis or protein phosphatase activity is suppressed. *J Cell Biol,* 179**,** 671-85.